# ImgEdit: A Unified Image Editing Dataset and Benchmark

**Yang Ye**[1,3,*], **Xianyi He**[1,3,*], **Zongjian Li**[1,3,*], **Bin Lin**[1,3,*], **Shenghai Yuan**[1,3,*],
**Zhiyuan Yan**[1,*], **Bohan Hou**[1], **Li Yuan**[1,2,†]

∗ Equal Contributors, † Corresponding Authors

[1] Peking University, Shenzhen Graduate School, [2] Peng Cheng Laboratory, [3] Rabbitpre AI
{yang.ye@stu, yuanli-ece@}.pku.edu.cn

## Abstract

Recent advancements in generative models have enabled high-fidelity text-to-image generation. However, open-source image-editing models still lag behind their proprietary counterparts, primarily due to limited high-quality data and insufficient benchmarks. To overcome these limitations, we introduce **ImgEdit**, a large-scale, high-quality image-editing dataset comprising one million carefully curated edit pairs, which contain both novel and complex single-turn edits, as well as challenging multi-turn tasks. To ensure the data quality, we employ a multi-stage pipeline that integrates a cutting-edge vision-language model, a detection model, a segmentation model, alongside task-specific in-painting procedures and strict post-processing. ImgEdit surpasses existing datasets in both task novelty and data quality. Using ImgEdit, we train **ImgEdit-E1**, an editing model using Vision Language Model to process the reference image and editing prompt, which outperforms existing open-source models on multiple tasks, highlighting the value of ImgEdit and model design. For comprehensive evaluation, we introduce **ImgEdit-Bench**, a benchmark designed to evaluate image editing performance in terms of instruction adherence, editing quality, and detail preservation. It includes a basic testsuite, a challenging single-turn suite, and a dedicated multi-turn suite. We evaluate both open-source and proprietary models, as well as ImgEdit-E1, providing deep analysis and actionable insights into the current behavior of image-editing models.[1]

## 1 Introduction

Recent progress in large-scale multi-modal datasets [43, 58, 45, 32, 63, 2] has enabled text-to-image models [56, 12, 13, 53, 14, 64, 22] to produce images with unprecedented fidelity. Among downstream applications, image editing [80, 74, 25, 27, 49, 60] stands out as especially important: users aim to apply precise local or global modifications to images, altering target regions while preserving the rest. The latest cutting-edge models, such as GPT-4o-Image [51] and Gemini-2.0-Flash [19] have recently achieved notable improvements in editing accuracy and instruction following ability. Additionally, these models have demonstrated a strong ability to perform complex, multi-turn edits, advancing image editing toward becoming a practical and powerful tool for real-world applications [77, 78, 37].

However, the performance gap between closed-source and open-source models continues to widen, primarily due to the lack of high-quality, publicly available editing datasets and accurate evaluation benchmarks in the open-source community. As shown in Table 1 and Table 2, the existing datasets [80, 85, 88, 59, 18, 28, 3, 81, 28] and benchmarks [85, 59, 71, 48] exhibit three main limitations: **(1) Sub-optimal data quality and prompt design**: Current collection pipelines typically begin with low-resolution images [43, 45, 31], generate prompts with open-source large language models [21, 66, 45, 42] that may introduce knowledge biases, synthesize edited image pairs using low-fidelity

---

[1]The source data are publicly available on https://huggingface.co/datasets/sysuyy/ImgEdit.

39th Conference on Neural Information Processing Systems (NeurIPS 2025) Track on Datasets and Benchmarks.

Table 1: **Comparison of Existing Datasets and ImgEdit.** "GPT score" denotes the quality score evaluated by GPT-4o [1]. "Fake score" quantifies the confidence of a forensic model [76] in identifying the edited image; lower values correspond to higher data quality. "Tiny-ROE ratio" denotes the ratio of samples whose edited region covers less than 1% ([85, 6, 28] do not utilize region-based in-painting, resulting in global changes.) "Concepts" counts the number of distinct words in prompts.

| Dataset | #Size | #Types | Res. (px)↑ | GPT Score↑ | Fake score↓ | Tiny-ROE(1%) Ratio↓ | Concepts↑ | ID Cons. Edit | Hybrid Edit | Multi-Turn |
|---|---|---|---|---|---|---|---|---|---|---|
| MagicBrush [85] | 10K | 5 | 500 | 3.88 | 0.987 | — | 2k | ✗ | ✗ | ✓ |
| InstructPix2Pix [6] | 313K | 4 | 512 | 3.87 | 0.987 | — | 11.6k | ✗ | ✗ | ✗ |
| HQ-Edit [28] | 197K | 6 | ≥ 768 | 4.55 | 0.186 | — | 3.7k | ✗ | ✗ | ✗ |
| SEED-Data-Edit [18] | 3.7M | 6 | 768 | 3.96 | 0.983 | 8% | 29.2k | ✗ | ✗ | ✓ |
| UltraEdit [88] | 4M | 9 | 512 | 4.25 | 0.993 | 9% | 3.7k | ✗ | ✗ | ✗ |
| AnyEdit [80] | 2.5M | 25 | 512 | 3.83 | 0.772 | 16% | 6.4k | ✓ | ✗ | ✗ |
| **ImgEdit** | 1.3M | 13 | ≥ 1280 | 4.71 | 0.050 | 0.8% | 8.7k | ✓ | ✓ | ✓ |

algorithms [56, 6], and apply coarse post-processing filters based on semantic scores [54, 7, 52] that measure editing quality poorly. Consequently, most datasets suffer from poor image resolution, simplistic prompts, negligible edit regions, inaccurate editing, concept imbalance, and imprecise filtering. **(2) Inadequate support for complex editing tasks**: Existing datasets rarely include edit types that (i) preserve identity consistency [51, 83], (ii) manipulate multiple objects simultaneously, or (iii) span multi-turn interactions [51, 19, 47]. Identity-preserving capability is critical for applications such as virtual try-on [75, 69, 84] and product design, whereas the latter two are indispensable in real-world scenarios and important for user experience. Although MagicBrush [85] and SEED-Data-Edit [18] contain multi-turn examples, they neglect the semantic relevance between prompts across different turns, which leads to failures in meeting the requirements for content understanding, content memory, or version backtracking. **(3) Limited benchmarking protocols**: The existing evaluation frameworks [71, 48, 30, 5] lack of diverse or reasonable evaluation dimensions. They do not stratify task difficulty, overemphasize the number of editing categories, and pay insufficient attention to evaluation dimensions or measurement accuracy. These limitations prevent current benchmarks from reliably characterizing the specific strengths and weaknesses of models [51, 19, 46, 80, 88, 85].

To address these challenges, we present **ImgEdit**, a unified framework that combines **(1) an automated** *data construction pipeline*, **(2) a large-scale editing** *dataset*, **an advanced editing** *model*, and **a comprehensive** *benchmark* **for evaluation**. As illustrated in Figure 2(left), we develop an automated pipeline to guarantee data quality. First, we discard images with low aesthetic scores [58], insufficient resolution, or negligible editable regions. Next, we generate object-level grounding annotations for the remaining images using an open vocabulary detector [8] and a visual segmentation model [57]. We then feed GPT-4o [1] with grounding information, target object, and specified edit type to generate a diverse set of single-turn and multi-turn prompts. Subsequently, task-specific workflows powered by state-of-the-art models [86, 13, 53] create the edited pairs. Finally, GPT-4o evaluates edit results and retains only those image pairs that follow the edit prompt while preserving visual fidelity. The resulting dataset consists of 1 million high-quality single-turn edit pairs covering 10 common editing operations, demonstrated in Figure 1 and Figure 3. These include a subset of object extraction tasks, wherein identity-consistent objects are isolated from complex scenes, as well as hybrid edit tasks involving instructions that reference multiple objects and editing operations. Additionally, the dataset contains 100,000 multi-turn interaction samples designed to include content understanding, content memory, and version backtracking edit tasks. To verify the effectiveness of the proposed dataset, we train **ImgEdit-E1** with ImgEdit, achieving new state-of-the-art performance across multiple image editing tasks. Finally, we propose **ImgEdit-Bench**, consisting of three key components: a basic editing suite that evaluates instruction adherence, editing quality, and detail preservation across a diverse range of tasks; an Understanding-Grounding-Editing (UGE) suite, which increases task complexity through challenging instructions (e.g., spatial reasoning and multi-object targets) and complex scenes such as multi-instance layouts or camouflaged objects; and a multi-turn editing suite, designed to assess content understanding, content memory, and version backtracking. To facilitate large-scale evaluation, we train **ImgEdit-Judge**, an evaluation model whose preferences closely align with human judgments. Our main contributions are summarized as follows:

**i) Robust Pipeline.** We introduce a high-quality data generation pipeline that ensures the dataset is diverse, representative, and of sufficient quality to support the development of image editing models.

**ii) New Dataset.** We construct *ImgEdit*, a large-scale, high-quality dataset comprising one million single-turn samples with ten representative edit tasks and 100,000 multi-turn samples containing three novel interaction types.

**iii) Reliable Benchmark.** We release *ImgEdit-Bench*, which evaluates models across tasks in three key dimensions, including a basic, challenging, and multi-turn suite.

**iv) Advanced Models.** We train *ImgEdit-E1* on ImgEdit, surpassing open-source models on many tasks. Moreover, we release *ImgEdit-Judge*, an evaluation model aligned with human preferences.

## 2 Related Work

### 2.1 Datasets for Image Editing

Table 1 compares representative instruction-driven image-editing datasets [6, 28, 85, 18, 88, 80]. InstructPix2Pix [6], EMU-Edit [59], HQ-Edit [28], and AnyEdit [80] rely almost entirely on synthetic or fully automated pipelines, whereas SEED-DataEdit [18], UltraEdit [88], MagicBrush [85] add varying degrees of human quality control. InstructPix2Pix is confined to the P2P [24] synthetic domain, hindering real-image transfer. MagicBrush improves real-world usability through high-quality human annotations but contains only 10,000 pairs. HQ-Edit enriches captions with GPT-4V [1] and DALL-E [61] yet produces images that limit realism. Recent datasets [80, 18] expand the range of edit types and dialog turns, but they are still confronted with limited data quality and insufficient prompt diversity. SEED-DataEdit introduces multi-turn interaction data [17, 44, 11, 9]; however, there is no interaction between each turn, rarely reflecting real-world workflows. Additionally, compositional operations in single-prompt and identity consistency edit remain under-represented.

### 2.2 Benchmarks for Image Editing

Current benchmarks [30, 48, 71, 85, 27, 59, 82, 87, 28, 80, 35, 36] for instruction-driven image-editing models [38, 20, 16, 70, 34, 15, 72] remain rudimentary. Earlier studies typically rely on generic similarity metrics—such as the CLIP score [54], PSNR [33], SSIM [73], —that correlate poorly with human judgments. MagicBrush [85] and EMU-Edit [59] broaden the scope of task-specific benchmarks, yet they still measure performance by similarity. SmartEdit [27] targets highly complex scenes but neglects most common settings. I2E-Bench [48] employs GPT-4o to produce human-aligned evaluations across diverse tasks; however, it employs a distinct metric for each task, which does not adequately capture the shared characteristics of editing. Moreover, none of the benchmarks differentiate between difficulty levels [10], which may result in unfair evaluations of the models. Although recent multimodal systems like GPT-4o-image [51] and Gemini-2.0-Flash [19] highlight **the need for multi-turn editing, no existing benchmark currently addresses this**, to our knowledge.

## 3 ImgEdit: A High Quality Dataset

**ImgEdit** provides high-fidelity edit pairs with accurate, comprehensive instructions, and encompasses a broader range of both practical and challenging edit types. Section 3.1 outlines the single- and multi-turn editing types, Section 3.2 details the data pipeline. We introduce **ImgEdit-E1** in Section 3.3, which is a cutting-edge edit model trained on ImgEdit. Section 3.4 presents the dataset statistics.

### 3.1 Edit Type Definition

We define two categories of editing tasks: single-turn and multi-turn. Single-turn tasks focus on covering comprehensive and practical tasks, whereas multi-turn tasks integrate interactions across instructions and images in continuous editing scenarios.

**Single-Turn Edit** Based on real-world editing practice, we divide single-turn tasks into four categories: local, global, visual, and hybrid edit, shown in Figure 1. **Local Edit** includes add, remove, replace, alter, motion change, and object extraction operations. Changes in color, material, or appearance are subsumed under alteration. Because editing human actions is a common use case [67], we also support motion changes specific to human subjects. Moreover, we introduce a novel object extraction task, for example—*"extract the cat to a white background"*—that isolates a specified subject on clean background while preserving identity consistency. This capability is valuable in many design pipelines [75] and is currently available only in GPT-4o-image [51]. **Global Edit** comprises background replacement and style or tone transfer. **Visual Edit** involves editing an image using a reference image. Given a reference object and an instruction, such as *"add a scarf*

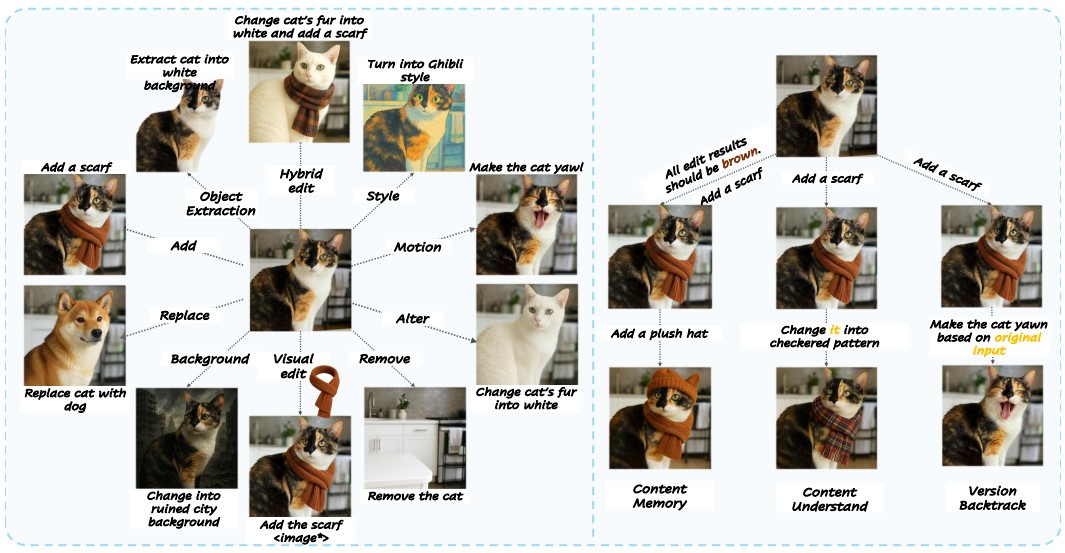

Figure 1: **Single- and Multi- Turn Edit Types.** (Left) Single-turn tasks include add, remove, replace, alter, background change, motion change, style, object extraction, visual edit, and hybrid edit. (Right) Multi-turn tasks include content memory, content understanding, and version backtracking.

*to the cat"*, the edit task performs the edit while ensuring object consistency. Unlike AnyEdit [80], we omit segment-, sketch-, and layout-guided variants since such visual cues are seldom supplied in practice. **Hybrid Edit** contains two local edit operations applied to several objects within a single instruction. They are created by randomly combining add, remove, replace, and alter operations—for example, *"add a scarf and change the cat's fur colour to white"*.

**Multi-Turn Edit**    Getting insights from existing multi-turn understanding benchmarks [62, 23, 89] and practical requirements, we identify three major challenges in multi-turn image editing, which are content memory, content understanding, and version backtracking, illustrated in Figure 1. **Content Memory** concerns global constraints introduced early in the dialogue. If the initial instruction stipulates that *"all generation must have a wooden texture"*, subsequent turns do not need to restate this requirement; however, the constraint must still be honored. **Content Understanding** refers to the ability to interpret later instructions that rely on pronouns or omitted subjects. For example, after the first instructs, *"Place a piece of clothing in the wardrobe on the left side of the image"*, second turn may request, *"Make it black"*, and third turn, *"Change into white"*, both implicitly referring to the clothing added in first turn. **Version Backtracking** denotes the capability to edit based on earlier versions of edit results. For example, *"Undo the previous change(or starting from the original input) . . ."* We believe that these three challenges cover most of the difficulties and distinguishing features of multi-turn interactive editing, which frequently arise in practical applications. The identity consistency issue [51] within dialogue, however, is better viewed as a single-turn generalisation problem and is unrelated to instruction comprehension in multi-turn interactions.

## 3.2    Automatic Dataset Pipeline

**Data Preparation**    We adopt LAION-Aesthetics [58] as our primary corpus. Compared to other datasets [43, 45], it offers greater diversity in scenes, higher resolution, and a more comprehensive range of object classes. We retain only images whose shorter side exceeds $1280$ pixels and whose aesthetic score [58] is above $4.75$, resulting in a $600k$ image subset. GPT-4o [1] is then used to regenerate concise captions and to extract editable objects and background nouns. Next, each candidate entity is localised with an open-vocabulary detector [8], and the resulting bounding boxes are refined into segmentation masks with SAM2 [57]. Every object and background region thus obtains both a bounding box and a mask. Because detection and segmentation are imperfect, we crop each object by its mask and compute (i) CLIPScore [54] between the crop and its object name, (ii) aesthetic quality [58], (iii) area ratio. Regions with low similarity, poor aesthetics, or negligible area are discarded, ensuring that the remaining targets are accurately identified and visually salient for subsequent editing. Specifically, we ensure that the edited area constitutes more than 40% of the

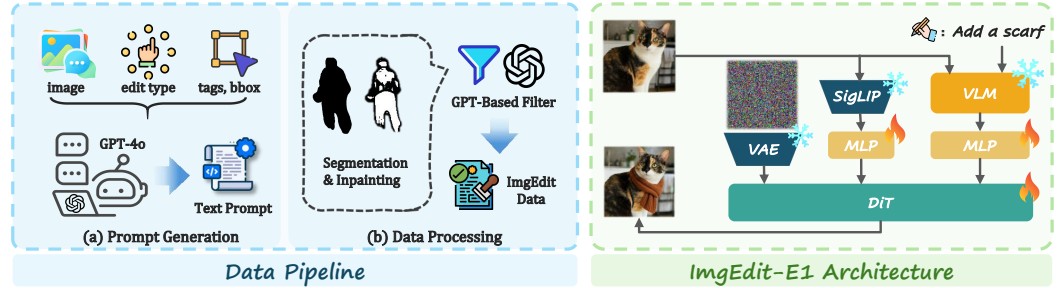

Figure 2: **The Data Pipeline and Model Architecture.** (Left) Our pipeline includes pre-filter, grounding and segmentation, caption generation, in-painting and post-processing, leveraging lots of state-of-the-arts models. (Right) ImgEdit-E1 includes a Qwen2.5-VL-7B [4] as text and image encoder, a SigLIP [68] provides low-level feature, and FLUX [13] as DiT backbone.

image for the background-changing task. For motion change edit, we additionally collect $150k$ image pairs from Open-Sora Plan [39] in-house videos that depict only people. Frames are temporally subsampled and their motions are annotated by GPT-4o [51], yielding the motion change subset.

**Instruction Generation**   We provide the original image caption, edit type, bounding box, and target object as conditioning information for prompt generation. Because precise localization of the target object is essential for successful editing, we instruct the language model to embed the position of object and approximate size in the editing instruction, using the bounding box as a reference. Less capable LLMs [66, 21] can introduce knowledge bias and produce low-quality prompts [80, 88]; therefore, we employ the state-of-the-art large language model [1], which not only understands diverse instruction formats and generates concept-rich editing instructions but also encodes spatial cues with high fidelity. For multi-turn prompt generation, we supply a few in-context examples and ask the model to produce the entire dialogue in a single pass; we then split the output into individual turns. To balance task complexity and operability, each dialogue is limited to three turns and is constructed from four basic operations: add, remove, replace, and alter.

**In-painting Workflow**   We select state-of-the-art generative models, such as FLUX [13] and SDXL [53], as base models. To achieve precise and controllable editing, we employ plug-ins, e.g., IP-Adapters [79], ControlNet [86], and Canny/Depth LoRA [26]. Based on these models and components, we construct data manufacturing pipelines tailored to each editing scenario. Within these pipelines, we incorporate novel techniques from the community to generate high-quality data. For instance, in reference-add or reference-replace tasks, we leverage the in-context capabilities of the FLUX architecture to maintain consistency and use FLUX-Redux to control semantics. Images produced by our method consistently outperform those in existing datasets [80, 88, 85, 18], exhibiting higher aesthetic quality and greater edit fidelity, as quantified in Section 3.4. In multi-turn dialogues, we reuse the same workflow, treating each request as an independent editing task.

**Post-Processing**   Since we have already performed a coarse filter during data preparation based on object area, CLIP score, and aesthetic score, we employ GPT-4o [1] to apply a precise filter in the post-processing stage. For each edit pair, GPT-4o assigns a quality score based on a prompt-guided rubric specific to the corresponding edit type. GPT-4o exhibits strong alignment with human preferences. Edit pairs that receive low scores are filtered out, resulting in a high-quality dataset after approximately $20\%$ of the candidates are discarded under this strict post-processing procedure.

### 3.3   ImgEdit-E1

To evaluate the quality of the collected data, we train **ImgEdit-E1** on ImgEdit. ImgEdit-E1 integrates a vision-language model (VLM) [4], a vision encoder [68], and a Diffusion-in-Transformer(DiT) backbone [13], as illustrated in Figure 2. The edit instruction and the original image are jointly fed into VLM, while the image is processed simultaneously by the vision encoder. The hidden states of VLM and the visual feature of the vision encoder are separately projected by MLPs and then concatenated, forming the text-branch input to DiT. Training proceeds in two stages [41], first optimizing MLPs and then jointly fine-tuning FLUX and MLPs.

### 3.4   Dataset Statistics

ImgEdit comprises 1 million high-quality image-editing pairs spanning 13 editing categories, including 100k multi-turn examples. Compared with existing datasets [80, 18, 88, 85, 28, 6], ImgEdit offers richer semantics, more detailed prompts, higher resolutions, greater editing accuracy, and overall superior visual fidelity. In particular, the object extraction and visual edit subsets constitute the first editing tasks with high subject consistency. The average short-side resolution of ImgEdit is 1280 pixels, whereas most competing corpora fall below this threshold. In terms of prompt diversity, ImgEdit contains 8,700 unique words. To assess editing accuracy, we randomly sampled 1,000 instances from each dataset and evaluated them with GPT-4o [51] according to Section 4, ImgEdit achieves the highest score. We further quantified the edited area for local edits tasks by pixel-wise differencing between the source and edited images; compared with other corpora, ImgEdit includes far fewer examples with small modification regions. Moreover, when a state-of-the-art editing-region detector [76] is applied, edits in ImgEdit are substantially harder to locate, indicating higher image quality. Comprehensive statistics are provided in Figure 3 and Table 1, with additional analyses and examples in the appendix.

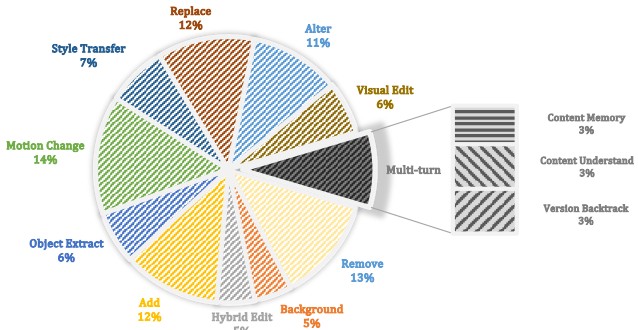

Figure 3: **Data Composition of ImgEdit**.

## 4 ImgEdit-Bench: A Comprehensive Benchmark

**ImgEdit-Bench** provides a comprehensive evaluation for both single- and multi-turn editing tasks. Section 4.1 outlines the composition of the benchmark dataset, Section 4.2 defines the evaluation metrics, and Section 4.3 introduces **ImgEdit-Judge**, a model to evaluate image editing tasks.

### 4.1 Benchmark Construction

We divide the capabilities of models into two categories: basic proficiency and performance in complex scenarios. The foundational evaluation measures the ability to complete easy tasks. The Understanding-Grounding-Editing (UGE) test suite assesses model capacity to perform understanding, grounding, and editing simultaneously within a single prompt. Finally, the multi-turn evaluation evaluates ability in content understanding, content memory, and version backtracking.

**Basic-Edit Suite**   Our benchmark comprises ten common image-editing tasks—add, remove, alter, replace, style transfer, background change, motion change, hybrid edit, and object extraction—evaluated on images that were manually collected from the Internet. To ensure semantic diversity, we select ten representative concepts from each of six super-categories (human, transportation, nature, animals, architecture, and necessities). For the add task, we pair each of ten relatively uncluttered background images with five prompts per concept. For the remove, alter, replace, cut-out, and hybrid-edit tasks, we chose photographs that contain few objects and a visually salient main subject. Style transfer is tested on popular styles; background change uses scenes suitable for substitution; motion change is assessed on human-centric images. All instructions are initially generated by GPT-4o [1] and then manually filtered. The resulting benchmark comprises 734 test cases with prompt lengths ranging from short to elaborate.

**Understanding-Grounding-Editing Suite**   We manually curated 50 complex images from the Internet that pose diverse challenges—partially occluded targets, scenes with multiple instances of the same category, camouflaged or visually inconspicuous objects, and uncommon editing subjects. For each image, we devised editing prompts that require spatial reasoning, multi-object coordination, compound or fine-grained operations, and large-scale modifications, thereby elevating the difficulty of comprehension, localization, and manipulation within a single prompt.

**Multi-Turn Suite**   For the multi-turn evaluation, we selected 30 images and manually designed prompts to emulate real-world use cases across three dimensions—content memory, content understanding, and version backtracking, resulting in 3 interaction rounds for each case.

Table 2: **Key Attributes of Open-source Edit Benchmarks.** The reliance of existing benchmarks on difficulty-level, multi-turn support, and evaluation metrics highlight the necessity of ImgEdit-Bench.

| Benchmark | #Size | #Sub-Tasks | Human Filtering | Difficult-Task Support | Multi-Turn Support | Metrics |
|---|---|---|---|---|---|---|
| EditBench [71] | 240 | 1 | ✗ | ✗ | ✗ | CLIP |
| EditVal [5] | 648 | 13 | ✓ | ✗ | ✗ | CLIP, VLM, manual |
| EmuEdit [59] | 3055 | 7 | ✗ | ✗ | ✗ | L1, CLIP, DINO |
| MagicBrush [85] | 1053 | 9 | ✓ | ✗ | ✗ | L1, L2, CLIP, DINO |
| AnyEdit [80] | 1250 | 25 | ✗ | ✗ | ✗ | L1, CLIP, DINO |
| I2EBench [48] | 2240 | 16 | ✓ | ✗ | ✗ | GPT(1 dim.) |
| ImgEdit-Bench | 779 | 14 | ✓ | ✓ | ✓ | GPT(3 dim.), Fake Detection |

## 4.2 Evaluation Metrics

We evaluated model performance along three dimensions: instruction adherence, image-editing quality, and detail preservation. Instruction adherence captures both prompt comprehension and conceptual understanding of the corresponding prompts. Because instruction adherence is fundamental to the editing task and cannot be fully separated from the other two aspects, the scores for image-editing quality and detail preservation are capped at the instruction-adherence score. Image-editing quality measures how precisely the target region is manipulated, whereas detail preservation measures the fidelity of regions that should remain unchanged. We employ the state-of-the-art Vision Language Model GPT-4o [1] to assign 1-5 ratings. For each task, we provide detailed scoring rubrics based on the three dimensions. In the multi-turn setting, human evaluators provided yes-or-no ratings for model output, following comprehensive guidelines designed to assess multi-turn capability. Additionally, we introduce a fake score to quantify how fake a generated image appears. To compute this, we use FakeShield [76], the latest open-source forensic detector that localizes editing artifacts within images. Specifically, we evaluate the recall (treating fake as the positive class) on various image-editing datasets and compare the results against ours. This allows us to assess and validate the visual realism and editing quality of our generated images.

## 4.3 ImgEdit-Judge

Because scores produced by vision–language models (VLMs) are more reasonable than traditional similarity metrics [54, 7], and no open-source VLM evaluator for image editing currently exists, we constructed a task-balanced and score-balanced corpus of $200k$ post-processed rating records and used it to fine-tune Qwen2.5-VL-7B [4]. We then performed a human study in which each image was rated by both human annotators, Qwen2.5VL-7B, ImgEdit-Judge, and GPT-4o-mini, and selected 60 images for detailed analysis. A judgment made by the model is considered correct when its score differs from the corresponding human score by no more than

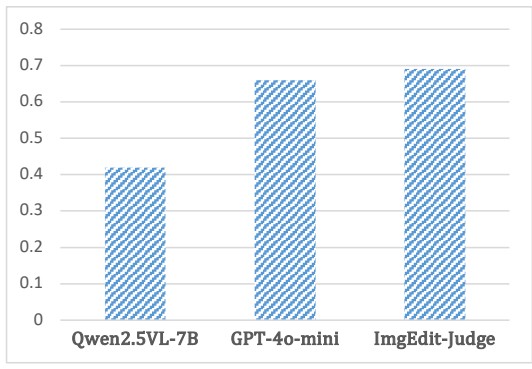

Figure 4: **Alignment Ratio with Human Preferences.**

one point. As illustrated in Figure 4, ImgEdit-Judge matches human judgments more closely than GPT-4o-mini and Qwen2.5-VL-7B, reaching almost $70\%$ agreement and surpassing the original Qwen2.5-VL by a substantial margin.

## 5 Experiments

In this section we conduct a comprehensive evaluation of existing editing models and ImgEdit-E1, Section 5.1 delineates the models under examination and experimental setup, Section 5.2 offers a qualitative and quantitative analysis of the results, and Section 5.3 presents further discussion.

## 5.1 Evaluation Setups

We run our single-turn benchmark on a wide range of image-editing models: the closed-source model includes GPT-4o-Image [51] (since Gemini-2.0-Flash [19] do not permit API request), while the open-source models include Step1X-Edit [46], Ultra-Edit [88], AnySD [80], MagicBrush [85], InstructPix2Pix [6], and ImgEdit-E1. Except for ImgEdit-E1 and Step1X-Edit, which use a Vision-Language Model as the text encoder and a Diffusion Transformer as the backbone, all other open-source models

rely on traditional UNet structures for diffusion models and pretrained text encoders [55, 54]. AnySD additionally incorporates a task-aware MoE block [40, 29]. All models are evaluated with identical prompts and images, and editing and evaluation are performed at the native resolution of each model. UltraEdit [88] and AnySD [80] generate outputs at $512 \times 512$ pixels, whereas the remaining models generate outputs at $1024 \times 1024$ pixels. Each experiment is repeated five times per model, and the mean score across the five runs is reported as the final result. We evaluate the only two models that support multi-turn editing: GPT-4o-Image and Gemini-2.0-Flash. We update the results of recent published models in Appendix D.2.

## 5.2 Evaluation Results

**Quantitative Evaluation** We first present a comprehensive qualitative evaluation of different methods, with results displayed in Figure 5.

Open-source models and closed-source models exhibit a significant performance gap, with GPT-4o-image [51] outperforming open-source models across all dimensions, only slightly lagging in some challenging tasks. Among the open-source models, ImgEdit-E1 and Step1X-Edit perform the best, achieving results close to closed-source models on a few tasks. ImgEdit-E1 demonstrates superior performance across all tasks, particularly excelling in object extraction and hybrid edit tasks due to its inclusion of high-quality data. This is reflected in its UGE suite scores, indi-

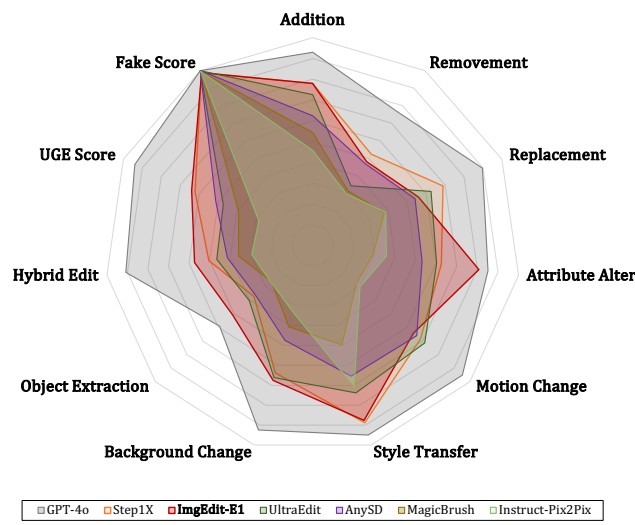

Figure 5: **Scores of Sub-Tasks for Each Model.**

cating stronger understanding, localization, and editing capabilities. Step1X-Edit exhibits similar performance to ImgEdit-E1 but falls short in background change, attribute alteration, and hard tasks. AnySD shows relatively average performance across various tasks but does not achieve outstanding results, possibly due to the broad range of editing tasks in its dataset, but lacks high-quality data. UltraEdit performs poorly in the removal task as it does not include this task in its dataset. MagicBrush and InstructPix2Pix suffer from issues such as image distortion and failure to follow instructions due to the limited quality and diversity of their training data and overly simple model structure. For all models, the editing outputs receive extremely high fake scores, indicating that detection models can still easily identify them. The specific scores of all models are provided in the appendix.

For multi-turn tasks, GPT-4o-Image exhibits strong version backtracking capabilities, whereas Gemini-2.0-Flash demonstrates minimal or no such ability. Models both possess content memory and content understanding capabilities; however, they may experience misunderstandings of some references or difficulty retaining premises in certain cases. More results are discussed in the appendix.

**Qualitative Evaluation** We select representative examples of diverse tasks for qualitative analysis, as shown in Figure 6. Only ImgEdit-E1 and GPT-4o-Image successfully preserve the snow on the bike while changing its color. In tasks involving object removal, AnySD and Step1X-Edit produce blurry results, Gemini incorrectly removes the street light together, and other models fail to follow instruction. In contrast, ImgEdit-E1 and GPT-4o-Image complete the task perfectly. ImgEdit-E1 and Step1X-Edit align most closely with the prompt in background change tasks among all open-source models. For replacement tasks, the results of closed-source models are noticeably more natural, while many open-source models fail to finish the edit. For attribute alteration tasks, only ImgEdit-E1 and the closed-source model accurately follow the instructions while preserving intricate details. Furthermore, only GPT-4o-Image and ImgEdit-E1 successfully perform the object extraction tasks.

## 5.3 Discussion

Based on our benchmark results, we identify three key factors that influence editing model performance: instruction understanding, grounding, and editing. **Understanding ability** is defined

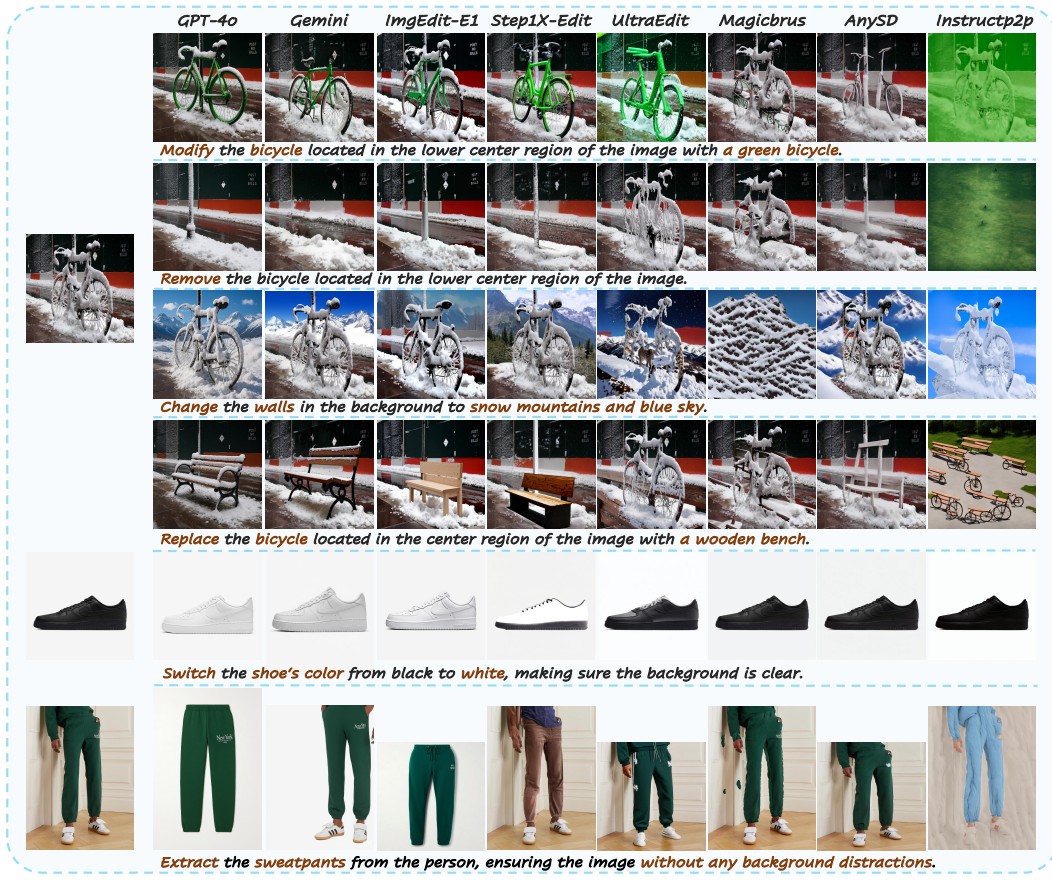

Figure 6: **Qualitative Comparison Among Different Editing Models**. ImgEdit-E1 surpasses all existing open-source models in instruction adherence, detail preservation, and visual quality, achieving results comparable to those of GPT-4o. Furthermore, owing to the novel editing tasks introduced in ImgEdit, it is capable of performing editing and extraction tasks while preserving identity consistency.

as the capacity of a model to comprehend editing instructions, which is largely determined by the text encoder and strongly affects editing performance. Conventional models using encoders such as T5 [55] or CLIP [54] manage simple tasks (e.g., style transfer) but perform poorly on complex, region-specific tasks. Our evaluations show that ImgEdit-E1 and Step1X-Edit substantially outperform other open-source models, underscoring the importance of stronger text encoders and more abundant text features. **Grounding ability** refers to the capacity to accurately identify and localize the specific region requiring editing, which is contingent upon both its ability to comprehend instructions and its visual perception capabilities. In our evaluations, ImgEdit-E1 exhibits superior performance compared to existing open-source editing models in tasks that demand precise localization, such as Attribute Alteration and Object Extraction, highlighting the importance of spatial information in prompts. **Editing ability** is the capacity to generalize across editing operations—depends chiefly on the quality, size, and diversity of the training datasets. The scarcity of high-quality data for Object Extraction yields poor performance on these tasks for other models, including GPT-4o [51], reaffirming the necessity of comprehensive, high-quality editing datasets.

## 6 Conclusion

This paper advances the image editing field by introducing **ImgEdit**, which overcomes data-quality limitations in existing datasets, introduces practical editing categories, and offers a robust pipeline for future dataset construction. Also, the strong performance of **ImgEdit-E1** validates the reliability of ImgEdit. Furthermore, **ImgEdit-Bench** evaluates models across novel dimensions, offering insights into data selection and architectural design for image-editing models. By providing *high-quality datasets, powerful editing methods, and comprehensive evaluation benchmarks*, we believe our

work helps close the gap between open-source approaches and SOTA closed-source models and drives progress across the entire field of image editing.

# 7 Acknowledgments

We thank all the anonymous reviewers for their constructive comments. This work was supported in part by the Natural Science Foundation of China (No. 62332002, 62202014, 62425101)

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

# NeurIPS Paper Appendix for ImgEdit: A Unified Image Editing Dataset and Benchmark

# A  More Discussion on ImgEdit-E1

## A.1  Details of Text Encoder

We conducted an experiment utilizing GPT-4o to generate detailed image descriptions, which were subsequently input into Flux to synthesize new images. Remarkably, despite Flux lacking access to the original images, the generated images exhibited a notable degree of similarity to the originals. Based on these findings, we posit that a text encoder with enhanced comprehension capabilities and extended context length would significantly improve editing tasks. To this end, we employed Qwen2.5-VL-7B as our text encoder, which supports a context length of up to 8k tokens with outstanding understanding ability. Additionally, Step1X-Edit also leveraged embeddings from the Vision-Language Model (VLM) as input for textual features, achieving superior results and thereby validating our hypothesis.

## A.2  Details of Vision Encoder

The primary distinction between ImgEdit-E1 and the Step1X-Edit model lies in the choice of vision encoder and how the vision features are utilized. In our model, we employed Siglip as the vision encoder, and its vision features were concatenated with the Vision-Language Model (VLM) features to serve as input for the text branch of Flux. In contrast, Step1X-Edit used a Variational Autoencoder (VAE) as the vision encoder, where the vision features were concatenated with noise to serve as input for the image branch. Our decision to use Siglip and its integration with text features was informed by the following findings: (1) The Flux-Redux model demonstrated that replacing text branch features with Siglip features enables basic control over the structural elements of an image. (2) OminiControl [65] also injecting low-level information using VAE, but it requires training a separate model for each task. Step1X-Edit, however, is unable to handle tasks such as generate canny or pose. This raises the question of whether VAE features introduce conflicts between different tasks, which remains unresolved. Based on these observations, we chose Siglip as the vision encoder to provide low-level information.

## A.3  Details of Generation Model

Flux is the current state-of-the-art in generative model, distinguished by its highly effective dual-stream architecture for integrating semantics and images. Moreover, its pre-trained weights substantially reduce training costs, making it an ideal foundation for our generative model.

## A.4  Details of Training Strategy

In the first stage of training, we freeze all parameters of Qwen2.5-VL and Flux, allowing only the MLP connecting Qwen2.5-VL to Flux to be trainable. The model is optimized using a global batch size of 128 and using prodigy [50], an adaptive optimizer with a learning rate set to 1.0. In the second stage, the Siglip encoder (Siglip v2-SO/16@512) is integrated to Flux using MLP, which is initialized from the pretrained Flux-Redux model. During this stage, the trainable parameters include the MLP connecting Siglipv2 [68] to Flux, the MLP connecting Qwen2.5VL to Flux, and the image branch of Flux. The second stage also employs the prodigy optimizer with a global batch size of 128. The model is trained for 50,000 steps in the first stage and 10,000 steps in the second stage.

## A.5  Details of Human Preference

We conducted an additional human evaluation to compare the capabilities of ImgEdit-E1 with those of other editing models, while also assessing the alignment between benchmark outcomes and human preferences. For this evaluation, we randomly selected five images from each subtask of ImgEditBench. For each original image and its corresponding editing instruction, edit results of each model were presented simultaneously to the evaluators. Considering the strong performance of gpt-4o-Image, we asked each evaluator to select the top two images from all candidates for each editing task. A total of ten evaluators participated in this assessment. The results are shown in Table 3. Although the difference is modest, ImgEdit is generally preferred over Step1x-Edit and significantly outperforms all other models except gpt-4o-Image.

## A.6  Limitations

As the core contribution of this paper is not ImgEdit-E1, we did not perform detailed ablation studies on its model structure, training data, or training process. The architecture of ImgEdit-E1 extends beyond that of a simple editing model; with additional training, it has the potential to evolve into a unified generative model capable of text-to-image generation, image editing, and low-level image

Table 3: **Top-2 Pick Ratio Comparison Across Different Models.** This table shows the percentage of cases where each model was selected as one of the top-2 performing models.

| Model | Top-2 Pick Ratio |
|---|---|
| Instruct-Pix2Pix | 0% |
| MagicBrush | 3% |
| AnySD | 5% |
| UltraEdit | 4% |
| ImgEdit-E1 | 51% |
| Step1X-Edit | 46% |
| GPT-4o-Image | 91% |

processing tasks(e.g., generating canny or depth image). However, while ImgEdit-E1's editing capabilities are not yet optimal for downstream scenarios such as text generation, this limitation is shared by all current open-source models. Future work will aim to address these limitations and explore the highlighted aspects in greater depth.

## B    More Details of ImgEdit Dataset

### B.1    Additional Ablation Studies

We designed ablation experiments to verify that our improvements are attributable to the data rather than the model architecture. Specifically, we kept the architecture of ImgEdit-E1 unchanged and ensured that all training hyperparameters remained consistent. During training, we replaced all ImgEdit data with AnyEdit data. The resulting model was then tested on ImgEditBench, the results are presented below. For comparison, we also included the results of AnySD.

Table 4: **Ablation Study: Data vs. Architecture.** This table compares the performance of ImgEdit-E1 trained on different datasets, demonstrating that the improvements are primarily due to data quality rather than model architecture.

| Sub-tasks | AnySD | ImgEdit-E1 | ImgEdit-E1 with AnyEdit |
|---|---|---|---|
| Remove | 2.23 | 2.40 | 1.53 |
| Background | 2.24 | 3.38 | 2.55 |
| Addition | 3.18 | 3.90 | 3.80 |
| Style | 2.85 | 4.38 | 3.66 |
| Adjust | 2.95 | 3.38 | 3.44 |
| Extract | 1.88 | 2.55 | 2.02 |
| Replace | 2.47 | 2.80 | 2.42 |
| Hybrid | 1.56 | 2.87 | 1.78 |
| Action | 2.65 | 3.21 | 2.67 |
| **Average** | **2.45** | **3.21** | **2.76** |

Table 5: **The detailed statistics of the ImgEdit dataset.**

| Type | #Image pairs |
|------|-------------|
| Add | 175467 |
| Remove | 160646 |
| Replace | 159395 |
| Alter | 135509 |
| Extraction | 59450 |
| Background | 44099 |
| Hybrid Edit | 28590 |
| Visual Edit | 59450 |
| Style | 64846 |
| Motion Change | 159008 |
| Content Memory | 30861 |
| Content Understand | 42139 |
| Version Backtrack | 42023 |

The experimental results clearly demonstrate the superiority of the ImgEdit dataset over the AnyEdit dataset, as evidenced by the comparison between ImgEdit-E1 and the ablation results. The substantial improvements observed in the extraction and hybrid tasks further support this advantage, given that the ImgEdit dataset contains a greater proportion of data relevant to these tasks than AnyEdit.

### B.2 Additional Details of Dataset Statistic

We list the word clouds for each task as follows 7:

For the selection of aesthetic score (4.75), we follow the official threshold provided by LAION, which can be found on LAION's website. We present detailed dataset statistics for all editing types in ImgEdit in Table 5. Post-processing results are also provided in the dataset, enabling users to apply additional filtering based on these results.

### B.3 Samples of Collected Data

Figure 8 presents various samples from the ImgEdit dataset, which consists of all edit tasks.

## C More Details of ImgEdit Pipeline

### C.1 Additional Details of Pre-Processing

To incorporate positional information into captions, we included the original image resolution and the size of the bounding box in the prompt. GPT generates instructions using the following metadata format: *"caption: {caption}, object: {object}, resolution: {resolution}, object bbox: {bbox}"* combined with task-specific prompts.

### C.2 Additional Details of In-painting Workflow

We developed an inpainting process tailored to each task by adopting the method from ComfyUI. This approach enables more effective utilization of advanced models within the community while offering a lightweight and user-friendly pipeline. The specific workflow diagram 9 is presented below.

**Replace & Background Change** Background change can be regarded as an extensive replacement task. Notably, we applied edge softening to the mask to mitigate abrupt transitions and ensure seamless editing effects.

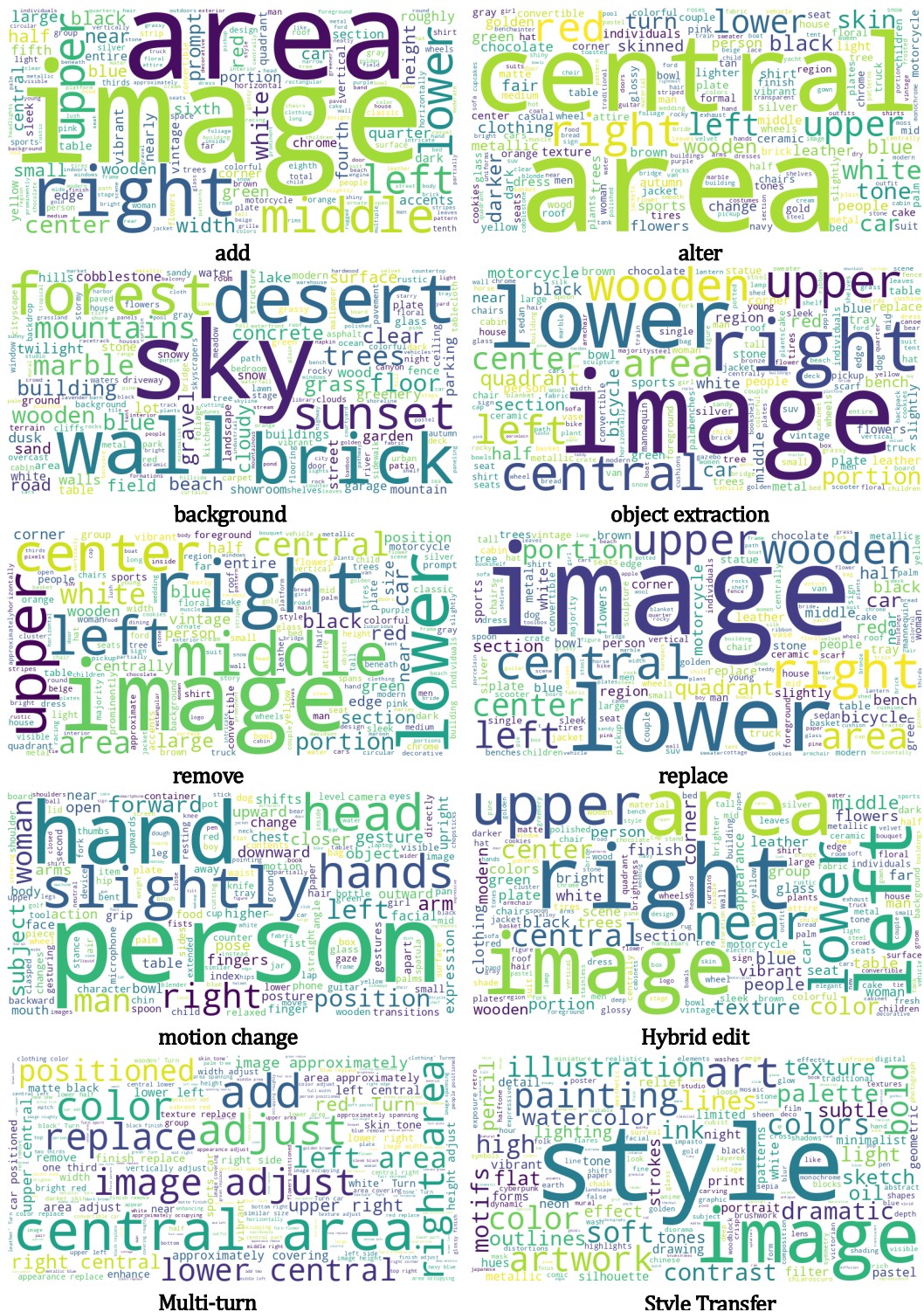

Figure 7: **Word cloud of different tasks.**

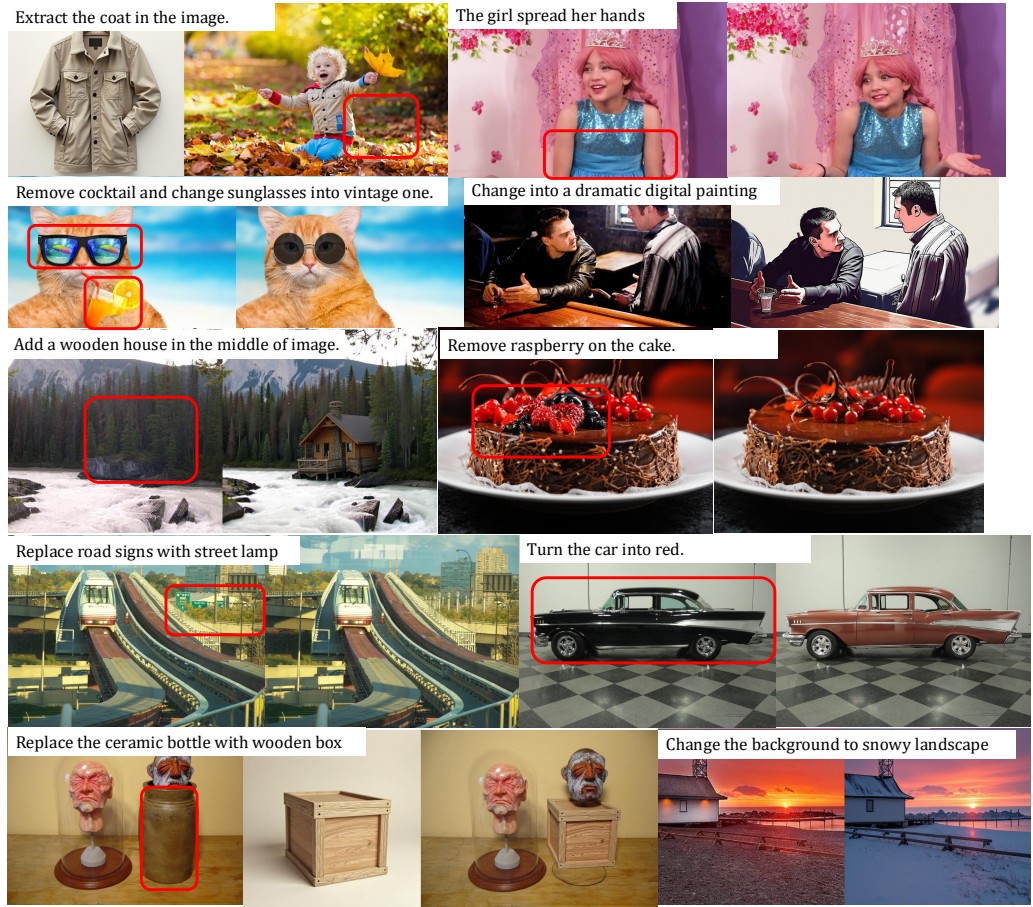

Figure 8: **Samples of Collected Data.**

**Add & Remove**   Add and Remove are inverse processes. We adopted a mask inpainting approach and designed prompts incorporating terms such as "empty" and "blank" to achieve the editing results.

**Alter**   For the attribute alteration task, we employed a Canny processor to generate edge maps as control signals. Additionally, we utilized Canny LoRA and ControlNet to perform the inpainting process, followed by softening the edges to enhance visual quality.

**Object Extraction & Visual Edit**   Object extraction and reference replacement are inverse processes. First, an object with a clean background is generated based on the prompt. Then, using Flux-Redux, the object in the real image is replaced with the generated object through in-context processing. The edited image and the object image are saved as use cases for object extraction, while the edited image, object image, and original image are saved as use cases for visual editing.

**Style Transfer**   For style transfer, we employed SDXL due to its superior fidelity in reproducing diverse styles. We also used Canny edge detection to ensure that the finer details of the image remained unchanged.

**Hybrid Edit**   The hybrid edit process followed the steps outlined above. This task was divided into two distinct editing iterations.

**Multi-Turn Edit**   The multi-turn edit process also followed the steps outlined above. It was divided into three editing iterations, each with path dependencies.

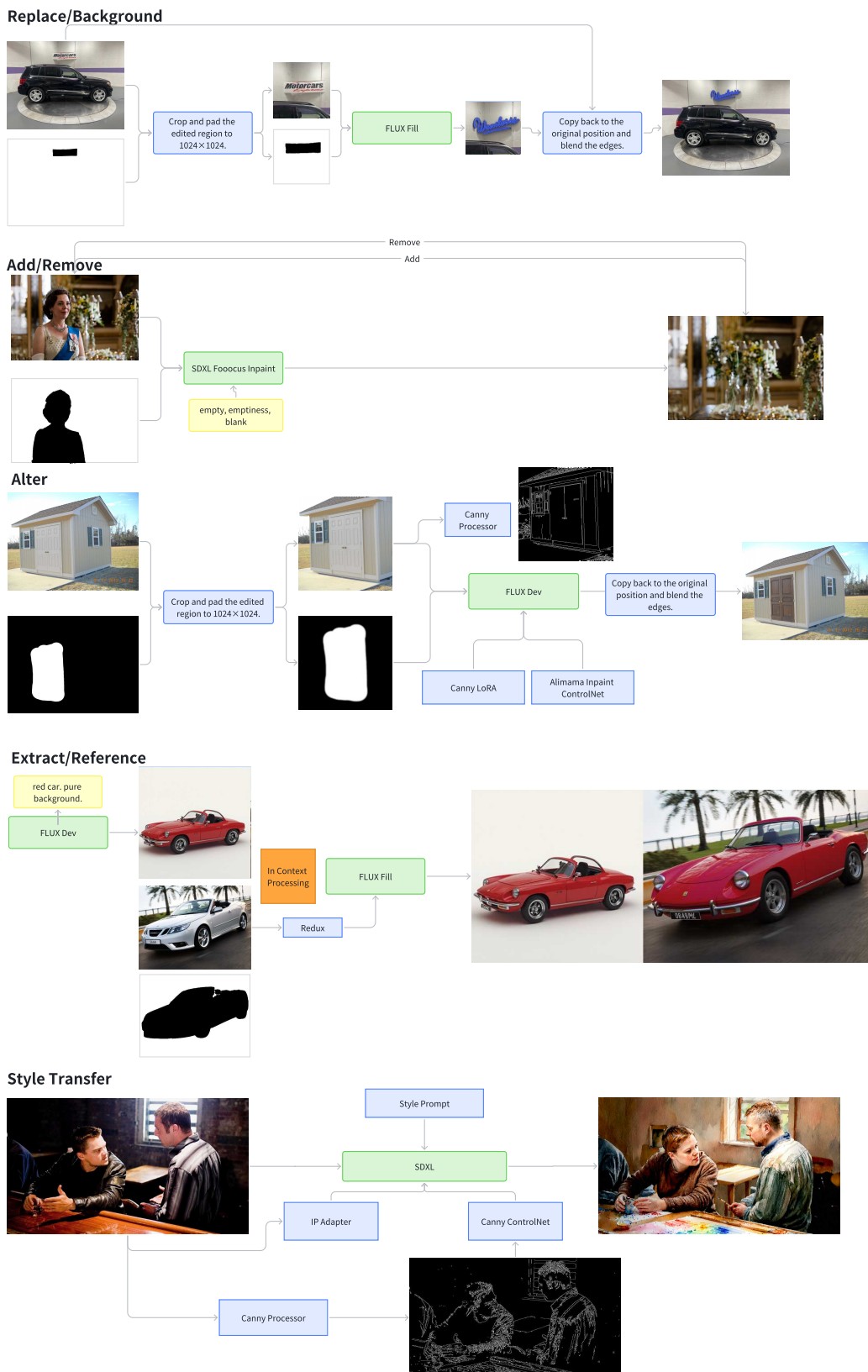

Figure 9: **In-paint Workflow.**

### C.3 Additional Details of Post-Processing

We utilized the original image, the resulting image, the general prompt, and the task-specific prompt as inputs, enabling GPT-4 to evaluate each pair by assigning a score. Examples of these evaluations are provided below.

### C.4 Limitations

As the in-painting process is mask-based, scenarios where the mask shape for the replacement object is significantly mismatched—such as attempting to replace a race car with a street lamp—can lead to replacement failures or highly unrealistic images. Most data of this nature are filtered out during post-processing. However, despite the use of state-of-the-art models in post-processing, a small subset of the processed data may still fail to align with human preferences.

## D    More Details of ImgEdit-Bench

**Instructpix2pix & MagicBrush & UltraEdit:**    This method extends a pretrained Stable Diffusion model into a image editor by fine-tuning its latent diffusion backbone on automatically generated triplets of (original image, edited image, edit instruction). During training, the network learns to denoise a latent corrupted by noise while simultaneously receiving the latent encoding of the source image and a text embedding of the desired edit. To achieve this, we augment the first convolutional layer with extra channels that concatenate the noisy latent and the source-image latent—these new weights are zero-initialized while all other parameters retain their pretrained values—and we repurpose the model's native text-conditioning mechanism to process edit instructions instead of captions. This lightweight adaptation harnesses the rich generative knowledge of the original model and enables it to perform high-fidelity, instruction-guided edits with minimal additional training.

**AnySD:**    First, a visual prompt projector takes CLIP-encoded image features and aligns them with text instructions, then injects this joint "visual prompt" into the UNet at each denoising step via cross-attention. Second, task-aware routing embeds Mixture-of-Experts blocks throughout the UNet: a lightweight router-informed by task embeddings—dynamically allocates each edit request to a subset of expert layers, so that different tasks invoke specialized attention pathways. Third, learnable task embeddings are inserted just before these MoE blocks to both shape the visual prompt and drive the decisions of router, ensuring that each editing operation fires at the right scale and scope.

**Step1X-Edit:**    The model fuses a multimedia large language model (MLLM), a lightweight connector, and a Diffusion Transformer (DiT) backbone to deliver precise, context-aware image edits. First, the user's edit instruction and reference image are jointly fed into a frozen MLLM (e.g. QwenVL) and become text feature input to our DiT network. In parallel, we average the MLLM's output tokens and project them into a global visual guidance vector, injecting rich semantic context into the diffusion process. To train the connector and DiT in tandem, they concat a noise-corrupted target image and a clean reference image into token sequences, and presenting this fused feature to the DiT. By initializing from pretrained Qwen and DiT weights and optimizing both modules jointly at a low learning rate, our method achieves high-fidelity, semantically aligned edits across a wide variety of user instructions.

### D.1 Discussion on evaluation Dimensions

The three dimensions defined in ImgEdit-Bench correspond to our analysis and are also related to model design. For instance, the ability for understanding is provided by more powerful VLMs, while low-level structural information (e.g., features from SigLip or VAE) ensures that the structure of the image remains unchanged, which corresponds to detailed preservation. Finally, the text encoder, vision encoder, and diffusion model collectively determine edit quality. The scores reflected by all the models demonstrate the combined effect of data and architecture. From qualitative and quantitative analyses, we observe that tasks such as removal, motion change, and extraction exhibit relatively poor performance across the board. Therefore, data related to these tasks should be given more emphasis, with an increased proportion in training. All models show particularly high fake scores, which may be due to the high sensitivity of detection models to AIGC-generated content. This also demonstrates that existing models are still unable to fully preserve non-edited regions without modification. Of course, the current field of detection models also needs to keep pace with the times and align more closely with human preferences.

Table 6: **Comparison results of different models on ImgEdit-Bench.** "Overall" is calculated by averaging all scores across tasks. We use GPT-4.1 for evaluation.

| Model | Add | Adjust | Extract | Replace | Remove | Background | Style | Hybrid | Action | Overall↑ |
|---|---|---|---|---|---|---|---|---|---|---|
| MagicBrush | 2.84 | 1.58 | 1.51 | 1.97 | 1.58 | 1.75 | 2.38 | 1.62 | 1.22 | 1.83 |
| Instruct-P2P | 2.45 | 1.83 | 1.44 | 2.01 | 1.50 | 1.44 | 3.55 | 1.20 | 1.46 | 1.88 |
| AnyEdit | 3.18 | 2.95 | 1.88 | 2.47 | 2.23 | 2.24 | 2.85 | 1.56 | 2.65 | 2.45 |
| UltraEdit | 3.44 | 2.81 | 2.13 | 2.96 | 1.45 | 2.83 | 3.76 | 1.91 | 2.98 | 2.70 |
| ICEdit | 3.58 | 3.39 | 1.73 | 3.15 | 2.93 | 3.08 | 3.84 | 2.04 | 3.68 | 3.05 |
| Step1X-Edit | 3.88 | 3.14 | 1.76 | 3.40 | 2.41 | 3.16 | 4.63 | 2.64 | 2.52 | 3.06 |
| UniWorld-V1 | 3.82 | 3.64 | 2.27 | 3.47 | 3.24 | 2.99 | 4.21 | 2.96 | 2.74 | 3.26 |
| BAGEL | 3.81 | 3.59 | 1.58 | 3.85 | 3.16 | 3.39 | 4.51 | 2.67 | 4.25 | 3.42 |
| OmniGen2 | 3.57 | 3.06 | 1.77 | 3.74 | 3.20 | 3.57 | 4.81 | 2.52 | 4.68 | 3.44 |
| Kontext-dev | 3.83 | 3.65 | 2.27 | 4.45 | 3.17 | 3.98 | 4.55 | 3.35 | 4.29 | 3.71 |
| Ovis-U1 | 3.99 | 3.73 | 2.66 | 4.38 | 4.15 | 4.05 | 4.86 | 3.43 | 4.68 | 3.97 |
| GPT-4o-Image | 4.61 | 4.33 | 2.90 | 4.35 | 3.66 | 4.57 | 4.93 | 3.96 | 4.89 | 4.20 |
| UniWorld-V2 | 4.29 | 4.44 | 4.32 | 4.69 | 4.72 | 4.41 | 4.91 | 3.83 | 4.83 | 4.49 |

## D.2 More Quantitative Analysis

We have listed the specific scores of the model below.

## D.3 Multi-Turn Qualitative Analysis

We list some test cases of gpt-4o-Image in Figure 10 and Gemini-2.5-flash in Figure 11.

## D.4 Limitations

Given the large number of test cases, relying solely on human evaluation is impractical. However, the evaluation results of GPT-4o is stable, there is no significant fluctuation between multiple tests. Furthermore, for multi-turn benchmarks, due to the limitations of GPT-4o's capabilities, we did not employ automated evaluation methods and instead opted for human evaluation.

# E More Details of ImgEdit-Judge

## E.1 Details of training Data

We train our ImgEdit-Judge on 1xNvidia 8A100 80G for a day. Our training data originates from a post-processing dataset. During the post-processing stage, GPT evaluates the data across three dimensions: instruction adherence, image-editing quality, and detail preservation. To address the unique requirements of each editing task, we designed task-specific prompts. For the final dataset, we ensured that the score distribution was nearly balanced to prevent bias and enable the model to handle diverse tasks effectively. Additionally, the dataset includes post-processed data from all task types.

## E.2 Details of Model and Hyper-parameters

We selected Qwen2.5-VL-7B as the base model due to its compatibility with single-GPU inference and its relatively strong visual capabilities. During fine-tuning, we enhanced the visual resolution of the model's vision encoder to accommodate the high-resolution image pairs in our training data, as detailed evaluation requires a vision encoder optimized for higher resolutions. To facilitate this, we unfroze the vision encoder, LLM, and MLP, and adopted the official hyperparameters provided by Qwen2.5 for training [4]. The model was trained for one epoch on our dataset, during which the loss decreased consistently without any significant fluctuations.

## E.3 Details of Human Evaluation Protocol

We doubled the size of our dataset to 120 images and conducted a new round of human evaluation in our revision. The details of our evaluation procedure remain unchanged from the original, and will be described below.

We recruited 10 evaluators to participate in the assessment. To ensure a balanced distribution of images rated 1 through 5, while also minimizing the workload for each evaluator, we instructed each participant to select 10 edited pairs for each score (from 1 to 5) out of 100 editing pairs generated by various models. The edit prompts and source images were randomly drawn from multiple data sources, such as AnyEdit, ImgEditBench, and others. From these selections, we constructed subsets

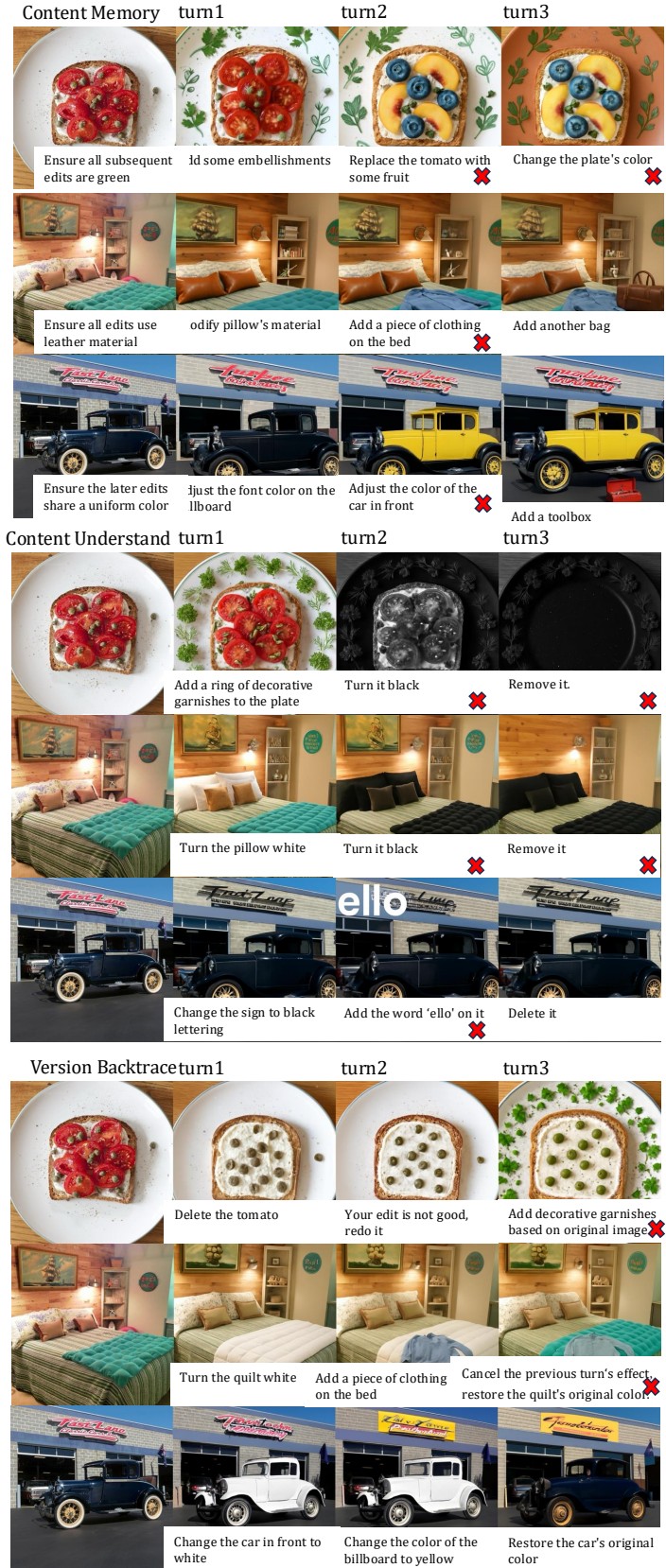

Figure 10: **Multi-Turn Cases of GPT-4o-Image.**

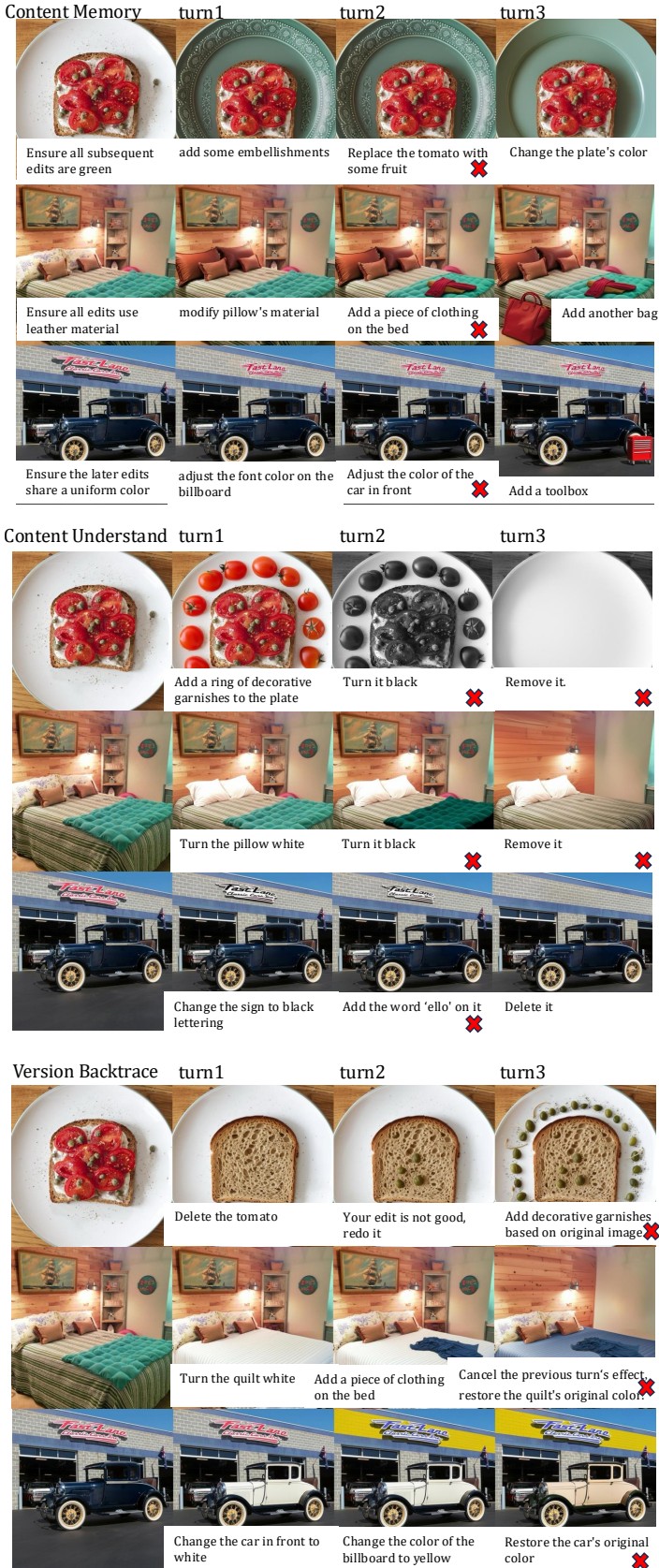

Figure 11: **Multi-Turn Cases of Gemini-2.5-flash.**

for each score by choosing edited pairs that received the same rating from at least two evaluators, resulting in 12 samples per score and a total of 60 samples in the final test set.

During evaluation, our prompt for LLMs adhered to the ImgEditBench scoring guidelines, scoring each edited pair based on instruction adherence, image-editing quality, and detail preservation. The prompts and input images were strictly controlled to be identical across all models. For each edited pair, we calculated the final model score by averaging scores from three dimensions.

To account for variations in human preferences and model outputs, we considered a model's score to be correct if it differed from the human evaluator's score by no more than one point. This evaluation protocol produced the result below.

Table 7: **Alignment Ratio with Human Preferences.** This table shows the percentage of cases where each model's score aligns with human evaluators' scores (within ±1 point).

| Model | Alignment Ratio |
|---|---|
| Qwen2.5VL-7B | 44.2% |
| GPT-4o-mini | 68.3% |
| ImgEdit-Judge | 71.6% |

## F  Additional Statement

### F.1  Errata

Due to the time limit and lack of money, we combination used is based on *gpt-4o-2024-11-20*, *gpt-4o-2024-08-06*, and *gpt-4.1-mini-2025-04-14* for post-processing. Since we did not update our prompt in time, about 1/5 of post-processing use one score to represent the edit quality, the other use three scores. So there maybe a little difference between the filter standard.

### F.2  Declaration of LLM Usage

We utilized Large Language Models (LLMs), such as ChatGPT, to support the preparation of this paper. Specifically, LLMs were employed for language-related tasks, including grammar correction, spelling checks, and word choice refinement, to improve the manuscript's clarity and fluency. Additionally, LLMs assisted with data processing and filtering (e.g., our ImgEdit-Bench uses GPT to score image-pairs), as well as generating draft figures to assist the authors in creating refined visualizations. All scientific content, analyses, and conclusions were independently conceived, validated, and interpreted by the authors.

### F.3  Potential Harms Caused by the Research Process

The images pairs of **ImgEdit** are derived from two open-source datasets —Laion-aes[58], and Open-Sora Plan [39]—that adhere to the MIT and Apache license. The licensing information for these data is explicitly stated on their respective platforms.

Data collection was made possible through the dedicated efforts of numerous contributors, including the authors of this paper and those involved in the manual evaluation. We consider individual hourly wages or compensation as personal information, and for privacy reasons, these details cannot be disclosed. Nonetheless, we can confirm that all participants have received appropriate compensation in accordance with the legal requirements of their respective countries or regions. The privacy of all participants is safeguarded, ensuring that no additional risks are posed to them.

## G  Social Impact and Potential Harmful Consequences

ImgEdit has developed datasets and corresponding benchmarks in the field of image editing to advance research in this domain. While image editing models have significant potential to enhance creativity, their broader societal impacts must be carefully evaluated during the development process.

**Environmental Resource Consumption**  Training image editing models demands substantial computational power, with a single large-scale training session potentially consuming tens of thousands

of kilowatt-hours of electricity—equivalent to the annual carbon emissions of several dozen cars. This high energy consumption exacerbates global climate change and consolidates computational resources within a few dominant tech companies, thereby deepening inequality within the research community. To mitigate these issues, efforts should focus on developing lightweight model architectures, optimizing distributed training efficiency, and promoting the adoption of green data centers powered by renewable energy to reduce the overall carbon footprint.

**Risks of Linguistic Homogenization and Cultural Bias**    Currently, ImgEdit's text prompts are limited to English, which may introduce biases in the model's ability to handle multilingual environments (e.g., Chinese). For example, when generating videos involving non-Western cultural symbols such as Hanfu or Kung Fu, a lack of relevant training data can lead to semantic distortions or cultural misunderstandings. Addressing this issue requires developing multilingual annotation systems, fostering open collaborative frameworks, and encouraging global researchers to contribute localized datasets to bridge language and cultural gaps.

**Ethical Concerns Related to Deepfake Misuse**    Image editing technologies are susceptible to misuse for malicious purposes, such as creating political disinformation, forging celebrity images, or fabricating criminal evidence. The realism achieved by these technologies already surpasses that of traditional Photoshop techniques, posing significant threats to public opinion, security, and judicial fairness. Effective countermeasures should integrate technical safeguards with regulatory oversight. These include embedding invisible watermarks into generative models, establishing blockchain-based content traceability protocols, and advocating for legislation mandating the labeling of AI-generated content. Public media literacy campaigns should also be launched to enhance societal resilience against misinformation.

## G.1    Impact Mitigation Measures

We are fully responsible for the authorization, distribution, and maintenance of **ImgEdit**. Our datasets and benchmarks are released under the CC-BY-4.0 license, while the code is released under the Apache license. All data is intended for academic research purposes to prevent misuse or improper use. All data are hosted on *GitHub* and *HuggingFace*, with the following links: https://github.com/PKU-YuanGroup/ImgEdit.

