# OpenReview forum: "ImgEdit: A Unified Image Editing Dataset and Benchmark"
_NeurIPS.cc/2025/Datasets_and_Benchmarks_Track — NeurIPS 2025 Datasets and Benchmarks Track poster_

### Official Review · Reviewer_bQft · 2025-06-29

**Rating:** 5
**Confidence:** 4

**Summary:**

The paper presents ImgEdit, a large-scale, high-quality dataset for instruction-based image editing. It consists of:

1. 1M single-turn edit samples and 100k multi-turn edit interactions.

2. A wide array of edit types: local (add, remove, replace), global (style, background), hybrid, and novel categories like object extraction and visual edit.

3. An automated, multi-stage pipeline leveraging SOTA models (e.g., GPT-4o, SAM2, ControlNet) for data generation, segmentation, inpainting, and filtering.

The paper also introduces:

1. ImgEdit-E1, a Vision-Language editing model trained on ImgEdit.

2. ImgEdit-Bench, a benchmark suite for evaluating image editing models across task types and difficulty levels.

3. ImgEdit-Judge, an automatic evaluator fine-tuned to align with human preferences.

**Additional Feedback:**

n/a

**Dataset Code Accessibility:**

Yes

**Dataset Code Comments:**

n/a

**Ethical Considerations:**

No, there are no or only very minor ethics concerns

**Final Justification:**

Thanks for the authors' time and effort on the rebuttal. I think this work is comprehensive, logical, and inspiring for the community, and I have raised my score to 5.

**Limitations Weaknesses:**

The paper proposed several good motivations, such as the editing fidelity, the interaction between each turn of editing, and the different difficulty levels of editing. However, I did not find the explicit demonstration of these. I list my questions and concerns in detail.

1. The image fidelity is an important issue in large-scale data generation. For example, the edited image should not edit the non-target regions. However, I did not find concrete quantitative results of comparison between the proposed dataset and previous datasets. I think it is better to show the accuracy of the masked regions in editing, especially for multi-turn editing. For example, in the series of PIE benchmarks [1, 2], the PSNR and LPIPS of the masked regions are reported, which can serve as a reference.

2. Following the above, I want to know how the benchmark deals with the editing related to pose and quantity, since these editing types will significantly change the layout of objects and images. How to evaluate such editing types is not clearly discussed or compared.

3. The paper does not evaluate train-free editing methods, which is actually a large group. Related methods include: RF-edit, P2P, etc. These methods generally tend to have better fidelity than the instruction-based methods. So, it is better to include these methods for a comprehensive evaluation.

4. If possible, I recommend making an ablation for the dataset components since no ablation studies are presented to isolate the impact of dataset components (e.g., instruction quality, object grounding accuracy). I think this study will be meaningful to know.

5. I find the supplementary is the same as the main submission. I actually look for the discussion about the user preference in alignment with the GPT and proposed Judge, which corresponds to section 4.3.

[1]. pnp inversion boosting diffusion-based editing with 3 lines of code.

[2]. ParallelEdits: Efficient Multi-Aspect Text-Driven Image Editing with Attention Grouping.

**Strengths Contributions:**

The paper discussed the motivations and advantages of the proposed benchmark over previous works in the related work. Concretely,

1. The motivation is clear and convincing. The current open-source image editing model has low image quality and limited editing scenarios. The high quality dataset supporting various editing types and scenarios is necessary and meaningful. It identifies concrete limitations in existing datasets (e.g., low quality, poor prompt diversity, lack of multi-turn edits), and convincingly argues why these hinder progress in open-source image editing.

2. The data generation pipeline is clearly presented as shown in Section 3.

3. The benchmarking is comprehensive. The paper includes a basic edit suite, a challenging Understanding-Grounding-Editing (UGE) suite, and a multi-turn suite with defined axes: instruction adherence, edit quality, and detail preservation.

4. The benchmark unifies several important points in the editing task, such as ID consistency, hybrid edit, and multi-turn edit.

---

> ### Author Rebuttal · Authors · 2025-07-31
>
> Thanks for your thorough comments and constructive suggestions. Your endorsement of our benchmark and dataset gives us significant encouragement. Here are our clarifications.
>
> > **Q1:** Image fidelity is crucial in large-scale data generation, especially ensuring non-target regions are not edited. The paper lacks quantitative comparisons with previous datasets. Reporting accuracy metrics for masked regions, as done in PIE benchmarks (e.g., PSNR and LPIPS), would strengthen the results, particularly for multi-turn editing.
>
> **A1:** Thank you for your question. Our approach adopts a "mask + inpainting" pipeline, and current generative models have demonstrated excellent consistency in preserving the surrounding regions during mask inpainting. Following your suggestion, we randomly selected a subset of 10,000 samples (excluding sub-tasks such as background and style editing, in which most regions are altered) and evaluated PSNR, SSIM, and LPIPS on the non-edited regions. Specifically, we computed these metrics only outside the bounding box corresponding to the edited area.
>
> We compared our results with VAE reconstruction and the data reported in the AnyEdit paper. As the AnyEdit dataset does not provide mask or bounding box information, we were unable to compute the same metrics directly and thus referenced the SSIM values reported in their paper. As shown in the table, **the consistency in the non-edited regions of our dataset is comparable to that of VAE reconstructions, demonstrating the high fidelity of our dataset in preserving unedited content.**
>
> | Data/Model    | PSNR ↑   | SSIM ↑   |  LPIPS↓  |
> | ------------- | -------- | -------- | ------   |
> | ImgEdit       | 27.84db  | 0.94     | 0.0433   |
> | SD-VAE[1]     | 30.19db  | 0.83     | 0.0568   |
> | AnyEdit[2]    |    -     | 0.66     |    -     |
>
> [1]Rombach R, Blattmann A, Lorenz D, et al. High-resolution image synthesis with latent diffusion models[C]//Proceedings of the IEEE/CVF conference on computer vision and pattern recognition. 2022: 10684-10695.
>
> [2]Yu Q, Chow W, Yue Z, et al. Anyedit: Mastering unified high-quality image editing for any idea[C]//Proceedings of the Computer Vision and Pattern Recognition Conference. 2025: 26125-26135.
>
> ---
>
> > **Q2:**  I want to know how the benchmark deals with the editing related to pose and quantity, since these editing types will significantly change the layout of objects and images. How to evaluate such editing types is not clearly discussed or compared.
>
> **A2:** Thank you for your question. Given the high diversity of editing tasks, we believe that traditional metrics such as PSNR often fail to provide reliable evaluations in many scenarios. Therefore, **we introduced GPT-based evaluation, in which GPT evaluates pairs of edited images according to the edit instructions and predefined criteria.** Specifically, we assessed model performance across three dimensions: instruction adherence, image-editing quality, and detail preservation. To this end, we employed the state-of-the-art vision-language model GPT-4o [1] to assign ratings on a scale from 1 to 5. A detailed description of our benchmarking methodology is provided in Section 4.2 of the manuscript, lines 254–268.
>
> Additionally, our benchmark dataset comprises nine common image-editing tasks: add, remove, alter, replace, style transfer, background change, motion change, hybrid edit, and extraction. The "pose" category you mentioned falls under motion change, while we do not include a benchmark specifically targeting quantity. The process of constructing our benchmark is described in detail in Section 4.1 of the manuscript, lines 227–252.
>
>
> ---
>
> > **Q3:** The paper does not evaluate train-free editing methods, which is actually a large group. Related methods include: RF-edit, P2P, etc. These methods generally tend to have better fidelity than the instruction-based methods. So, it is better to include these methods for a comprehensive evaluation.
>
>
> **A3:** Thank you for your meaningful question. Following your suggestion, **we conducted ImgEditBench evaluations on the RF-edit[1] model**, using parameter settings consistent with the official implementation. The results of the evaluation are presented below, we added ImgEdit-E1's score for comparison.
>
>
> | Model    | Remove | Background | Addition | Style | Adjust | Extract | Replace | Hybrid | Action | **average** |
> |-------------|--------|------------|----------|-------|--------|---------|---------|--------|--------|-------------|
> | RF-edit[1]  |  1.08  |   1.68     |   3.03   | 2.42  |  1.92  |  2.19   |  1.82   |  1.33  | 2.37   | **2.02**    |
> | ImgEdit-E1  |  2.40  |   3.38     |   3.90   | 4.38  |  3.38  |  2.55   |  2.80   |  2.87  | 3.21   | **3.21**    |
>
> While these kinds of training-free methods demonstrate excellent performance in image and video inversion as well as certain specific applications, they currently struggle to handle complex edit instructions effectively. We will update our revision to include our results, as well as a discussion and comparison of RF-edit and similar training-free approaches. Additionally, we will continue to monitor developments in training-free methods and incorporate timely evaluations as new techniques emerge.
>
>
> [1] Wang J, Pu J, Qi Z, et al. Taming rectified flow for inversion and editing[J]. arXiv preprint arXiv:2411.04746, 2024.
>
>
> ---
>
> > **Q4:** If possible, I recommend making an ablation for the dataset components since no ablation studies are presented to isolate the impact of dataset components (e.g., instruction quality, object grounding accuracy). I think this study will be meaningful to know.
>
>
> **A4:** Thank you for your insightful question. However, we believe that conducting an ablation study in this context is both challenging and resource-intensive, and we were unable to reconstruct the dataset and train multiple models for ablation within the rebuttal period.
>
> **We believe that factors such as instruction quality and object grounding accuracy are intuitively important, and we have taken them into careful consideration during the dataset design process,** striving to achieve the highest possible standards. As a result, these efforts have likely contributed to the superior quality of our ImgEdit dataset.
>
> ---
>
> > **Q5:** I actually look for the discussion about the user preference in alignment with the GPT and proposed Judge, which corresponds to section 4.3.
>
> **A5:** Thank you for your question. The appendix is concatenated after the main submission in supplementary. We would like to clarify our evaluation protocol as follows (also see Appendix E.3):
>
> We recruited 10 evaluators to participate in the assessment. To ensure a balanced distribution of images rated 1 through 5, while also minimizing the workload for each evaluator, we instructed each participant to select 10 edited pairs for each score (from 1 to 5) out of 100 editing pairs generated by various models. The edit prompts and source images were randomly drawn from multiple data sources, such as AnyEdit, ImgEditBench, and others. From these selections, we constructed subsets for each score by choosing edited pairs that received the same rating from at least two evaluators, resulting in 12 samples per score and a total of 60 samples in the final test set.
>
> During evaluation, our prompt for LLMs adhered to the ImgEditBench scoring guidelines, scoring each edited pair based on instruction adherence, image-editing quality, and detail preservation. The prompts and input images were strictly controlled to be identical across all models. For each edited pair, we calculated the final model score by averaging scores from three dimensions.
>
> To account for variations in human preferences and model outputs, we considered a model’s score to be correct if it differed from the human evaluator’s score by no more than one point. This evaluation protocol produced the bar chart shown in Figure 4.

---

> > ### Author Response · Authors · 2025-08-01
> >
> > Dear Reviewer bQft,
> >
> > Thank you very much for your positive feedback and encouragement. We are truly grateful for your support. It is wonderful to know that our response has resolved your concerns.
> >
> > Best regards,
> >
> > On behalf of all ImgEdit authors

---

### Official Review · Reviewer_YkpG · 2025-07-02

**Ethics Flags:** Data privacy, copyright, and consent
**Rating:** 4
**Confidence:** 2

**Summary:**

This paper proposes a new dataset, ImgEdit, a trained model, ImgEdit-E1, and an evaluation benchmark, ImgEdit-Bench, for the image editing task. ImgEdit is a novel dataset because it includes both single-turn edits and multi-turn edits. To evaluate image editing models from diverse dimensions, the paper proposes a novel benchmark composed of three key dimensions.

**Dataset Code Accessibility:**

Yes

**Ethical Comments:**

Some images for the benchmark were downloaded from the Internet (L235, 244). Have any copyright issues been considered?

**Ethical Considerations:**

Yes, there are ethics concerns that require attention by the authors

**Final Justification:**

As described in "Response to the authors", all of my concerns are solved. I raise my score 3 -> 4.

**Limitations Weaknesses:**

- Some detailed information is missing in the dataset pipeline. For example, in L152, why was 4.75 employed for the threshold for the aesthetic score?
- The manuscript does not provide details on the hyperparameters for training ImgEdit-E1.
- In section 5.2, the authors compared ImgEdit-E1 with other models. However, to evaluate the effectiveness of the ImgEdit dataset, training a model with other datasets and comparing it with them is required. Without this experiment, it cannot be conclusively stated that the improvement in model performance is due to the dataset.

**Strengths Contributions:**

- The proposed dataset, ImgEdit, is large and includes both single-turn and multi-turn edit data.
- The proposed benchmark is novel. Compared with previous benchmarks, the proposed one covers many scenarios. From a user study, it is experimentally shown that the proposed evaluation model is aligned with human preferences.

---

> ### Author Rebuttal · Authors · 2025-07-31
>
> Thank you for the time, thorough comments, and nice suggestions. We are pleased that you acknowledged the large scale dataset and novelty of benchmark. we are pleased to clarify your questions step-by-step.
>
>
> > **Q1:** Some detailed information is missing in the dataset pipeline. For example, in L152, why was 4.75 employed for the threshold for the aesthetic score?
>
> **A1:** Thank you for your question. We would like to further clarify some details of our pipeline. For the selection of aesthetic evaluation, we followed the official threshold provided by LAION, which can be found on LAION's website. Additional details related to the data pipeline, such as the inpainting workflow, prompts, and sampling procedures, are provided in the supplementary materials (Appendix C, in the supplementary material, Pages 22–24, lines 816-848).
>
> We will also include more detailed guidance referencing the appendix in the revised manuscript.
>
>
> ---
>
> > **Q2:** The manuscript does not provide details on the hyperparameters for training ImgEdit-E1.
>
> **A2:**
> Thank you for your question. We would like to further clarify some details of our training.
>
> The model architecture, hyperparameters, and training details of our model are provided in Appendix A of the supplementary materials (Pages 20, lines 761-806). We list them below.
>
> In the first stage of training, we freeze all parameters of Qwen2.5-VL and Flux, allowing only the MLP that connects Qwen2.5-VL to Flux to be trainable. The model is optimized using a global batch size of 128 and Prodigy [1], an adaptive optimizer with a learning rate set to 1.0. In the second stage, the Siglip encoder (Siglip v2-SO/16@512) is integrated to Flux using MLP, which is initialized from the pre-trained Flux-Redux model. During this stage, the trainable parameters include the MLP connecting Siglipv2 [2] to Flux, the MLP connecting Qwen2.5-VL to Flux, and the image branch of Flux. The second stage also employs the Prodigy optimizer with a global batch size of 128. The model is trained for 50,000 steps in the first stage and 10,000 steps in the second stage.
>
> We will also include more detailed guidance referencing the appendix in the revised manuscript.
>
> [1] Mishchenko K, Defazio A. Prodigy: An expeditiously adaptive parameter-free learner[J]. arXiv preprint arXiv:2306.06101, 2023.
>
> [2] Tschannen M, Gritsenko A, Wang X, et al. Siglip 2: Multilingual vision-language encoders with improved semantic understanding, localization, and dense features[J]. arXiv preprint arXiv:2502.14786, 2025.
>
> ---
>
> > **Q3:** In section 5.2, the authors compared ImgEdit-E1 with other models. However, to evaluate the effectiveness of the ImgEdit dataset, training a model with other datasets and comparing it with them is required. Without this experiment, it cannot be conclusively stated that the improvement in model performance is due to the dataset.
>
>
> **A3:** Thank you for your insightful question. Following your suggestion, **we designed ablation experiments to verify that our improvements are attributable to the data rather than the model architecture.** Specifically, we kept the architecture of ImgEdit-E1 unchanged and ensured that all training hyperparameters remained consistent; detailed settings can be found in lines 789–797 of the Appendix. During training, we replaced all ImgEdit data with AnyEdit [1] data. The resulting model was then tested on ImgEditBench, the results are presented below. For comparison, we also included the results of AnySD.
>
> | sub-tasks  | AnySD | ImgEdit-E1 | ImgEdit-E1 with AnyEdit |
> |-------|----|-------|---------|
> | Remove | 2.23  | 2.40| 1.53  |
> | Background | 2.24  | 3.38| 2.55 |
> | Addition| 3.18  | 3.90 | 3.80  |
> | Style | 2.85  | 4.38  | 3.66  |
> | Adjust | 2.95  | 3.38 | 3.44 |
> | Extract  | 1.88  | 2.55| 2.02 |
> | Replace | 2.47  | 2.80  | 2.42|
> | Hybrid  | 1.56  | 2.87 | 1.78 |
> | Action  | 2.65  | 3.21 | 2.67 |
> |  **average**  | **2.45**  |  **3.21**  |  **2.76↓** |
>
> **The experimental results clearly demonstrate the superiority of the ImgEdit dataset over the AnyEdit dataset**, as evidenced by the comparison between ImgEdit-E1 and the ablation results. The substantial improvements observed in the extraction and hybrid tasks further support this advantage, given that the ImgEdit dataset contains a greater proportion of data relevant to these tasks than AnyEdit.
>
> We hope our experiments address your concerns. We will update the revision with the latest experimental results.
>
> [1] Yu Q, Chow W, Yue Z, et al. Anyedit: Mastering unified high-quality image editing for any idea[C]//Proceedings of the Computer Vision and Pattern Recognition Conference. 2025: 26125-26135.
>
>
> ---
>
> > **Q4:** Ethical Comments: Some images for the benchmark were downloaded from the Internet (L235, 244). Have any copyright issues been considered?
>
> **A4:** Thank you for your reminder. Yes, we have considered copyright issues in Appendix G. The ImgEditBench data originate from websites that are licensed under Creative Commons Attribution 4.0 (CC BY 4.0) and Creative Commons Zero (CC0).

---

> > ### Comment · Reviewer_YkpG · 2025-08-01
> > **Response to the authors**
> >
> > Dear authors
> >
> > Thank you for addressing all of my concerns.
> >
> > I understand that some missing information will be contained in the revised manuscript or already included in appendix (Q1, 2, and 4).
> > I appreciate authors for providing ablation experiments for addressing Q3. I understand that training a model with the proposed ImageEdit dataset is efficient.
> >
> > Since all of my concerns are solved, I will raise my score.

---

> ### Author Response · Authors · 2025-08-01
>
> Dear Reviewer YkpG,
>
> Thank you for your encouraging feedback and support. We are delighted that our clarifications have met your expectations. Your constructive attitude is greatly valued !
>
>
> With appreciation,
>
> On behalf of all ImgEdit authors

---

### Official Review · Reviewer_WnbF · 2025-07-02

**Rating:** 6
**Confidence:** 4

**Summary:**

This paper introduces ImgEdit, a comprehensive framework comprising a large-scale dataset, a benchmark, and associated models for instruction-based image editing. The paper contributes a benchmark, a dataset, and an evaluation model.

**Dataset Code Accessibility:**

Yes

**Ethical Considerations:**

No, there are no or only very minor ethics concerns

**Final Justification:**

Thanks authors responce, I remain my recommendation.

**Limitations Weaknesses:**

1. Reliability of ImgEdit-Judge:
The human study to validate ImgEdit-Judge is based on only "60 images for detailed analysis (in L280-281)" .This sample size is far too small to make robust claims about the alignment of ImgEdit-Judge with human preferences across the vast and diverse space of image editing tasks. The process of selecting these 60 images is not detailed, leaving open the possibility of selection bias. A more extensive and carefully designed human study is needed to substantiate the claims about the reliability of the proposed automated evaluation metrics.

2. Experimental results:
The paper reports a crucial finding in Section 5.2: for all models, the editing outputs have extremely high fake scores, indicating that detection models can still easily identify them. This suggests that even models trained on the high-quality ImgEdit data still produce non-authentic images. But it is mentioned briefly, I think the authors should explore these results in depth. Especially, it seems to conflict with the low "fake score" reported for the ImgEdit dataset itself in Table 1. This tension needs to be discussed: why does training on "highly real" data still lead to "highly fake" outputs?

Although having the above weakness, the paper still proposes a powerful evaluation suite for instruction-based image editing, so I recommend accepting.

**Strengths Contributions:**

This paper has great contributions to the image editing community, as its dataset supports single- and multi-turn editing. And all suites, including the data pipeline, the dataset, and the judge model, are publicly available.

---

> ### Author Rebuttal · Authors · 2025-07-31
>
> Thanks for your time and the constructive suggestions. Your recognition of the paper's high contribution is greatly appreciated. Here are additional responses and clarifications based on your comments.
>
>
> > **Q1:** Reliability of ImgEdit-Judge: A more extensive and carefully designed human study is needed to substantiate the claims about the reliability of the proposed automated evaluation metrics.
>
> **A1:**
> Thank you for your suggestion. Following your advice and considering time and resource constraints, **we doubled the size of our dataset to 120 images and conducted a new round of human evaluation.** The details of our evaluation procedure remain unchanged from the original, and will be described below. (also mentioned in Supplementary Appendix E.3):
>
> We recruited 10 evaluators to participate in the assessment. To ensure a balanced distribution of images rated 1 through 5, while also minimizing the workload for each evaluator, we instructed each participant to select 10 edited pairs for each score (from 1 to 5) out of 100 editing pairs generated by various models. The edit prompts and source images were randomly drawn from multiple data sources, such as AnyEdit, ImgEditBench, and others. From these selections, we constructed subsets for each score by choosing edited pairs that received the same rating from at least two evaluators, resulting in 12 samples per score and a total of 60 samples in the final test set.
>
> During evaluation, our prompt for LLMs adhered to the ImgEditBench scoring guidelines, scoring each edited pair based on instruction adherence, image-editing quality, and detail preservation. The prompts and input images were strictly controlled to be identical across all models. For each edited pair, we calculated the final model score by averaging scores from three dimensions.
>
> To account for variations in human preferences and model outputs, we considered a model’s score to be correct if it differed from the human evaluator’s score by no more than one point. This evaluation protocol produced the result below.
>
>
> |                   | Qwen2.5VL-7B   |GPT-4o-mini|  ImgEdit-Judge|
> | --------          | -------- | --------  | --------      |
> | alignment ratio     |   44.2%  | 68.3%     |    71.6%      |
>
>
>
> ---
>
> > **Q2:** The paper finds that editing outputs have high fake scores, which appears to conflict with the low fake score of the ImgEdit dataset itself in Table 1. The authors should discuss why training on realistic data still produces easily detected fake outputs.
>
> **A2:** Thank you for your insightful question. **We incorporated the fake detection score based on the intuitive perception that, although gpt-4o-Image produces high-quality edits, its results can sometimes appear unnatural.** Currently, no existing benchmark addresses this aspect.
>
> However, we found that the detector’s sensitivity far surpasses that of human evaluators; it is able to identify AIGC images by low-level information, even when the images appear realistic to the human eye. As a result, the current fake detection score may not be fully adequate. Nonetheless, we would like to point out that **"realism of edits" is an important dimension that could further advance the capabilities of editing models.** Your suggestion has inspired us to pursue the development of a more human-aligned fake detector as part of our future work.
>
>
> We attribute the low fake score of our dataset and the comparatively higher fake score of ImgEdit-E1’s edited results to the following reasons: Our data pipeline is designed to edit only the masked regions of an image, meaning that the majority of each image remains sourced from real photographs.  In contrast, for ImgEdit-E1, the generative model aims to approximate the complex distribution of real images. However, this process tends to favor producing "average" results, often neglecting the subtle texture details present in genuine data and resulting in the commonly observed "AI smoothing effect". This phenomenon is an inherent bias of generative models, rather than a limitation of our dataset. Additionally, ImgEdit-E1 employs SigLIP as the image encoder, which emphasizes extraction of high-level semantic features. This can further reduce the perceived realism of edited images. In fact, most current generative models are trained on real data yet remain detectable as synthetic.

---

> ### Comment · Area_Chair_uy6T · 2025-08-05
>
> Dear reviewer,
> please read the other reviews and the author response, and start a discussion with the authors promptly to allow time for an exchange.
> Your AC

---

### Official Review · Reviewer_vFmQ · 2025-07-05

**Rating:** 5
**Confidence:** 4

**Summary:**

The paper introduces **ImgEdit**, a large-scale image editing dataset and evaluation benchmark aimed at bridging the performance gap between open-source and cutting-edge proprietary image editing models. The authors’ objectives are to provide high-quality training data for image editing tasks and a comprehensive benchmark to evaluate models on instruction-based editing.

**Dataset Code Accessibility:**

Yes

**Ethical Considerations:**

No, there are no or only very minor ethics concerns

**Final Justification:**

I'd like to raise my rate from 4 to 5. Here's the justifications:

1. The authors conducted analyses regarding potential biases in the stages of image generation, prompt generation, and filtering. Even biases stem from cumulative effects (original image content, segmentation models, LLM prompts), making them hard to avoid entirely, but mitigation strategies are applied or proposed.

2. The authors fine-tuned ​​ImgEdit-E1​​ on their multi-turn dataset, testing basic multi-turn editing with user study provided.

3. ImgEdit-E1 is ​​human-preferred​​ over comparison methods, though gpt-4o-Image remains the strongest. Results will be added to the revision.

4. ImgEdit-E1's gains come ​​mainly from its new dataset​​, not architecture changes. Step1X-Edit comparison invalid due to proprietary data.

**Limitations Weaknesses:**

1. **All edited images in ImgEdit are synthetic**, produced by diffusion models and then filtered. This means the dataset’s diversity and realism are ultimately bounded by the capabilities and biases of those generative models. If the diffusion models struggle with certain edit types or object categories, the resulting dataset may under-represent those cases. The authors partially mitigate this by using high-quality generation models and filtering out obvious failures, but **some subtle artifacts or biases could remain**. Did you observe any bias from using GPT-4 for prompt generation and filtering? For example, are certain styles or objects over-represented because of GPT-4’s choices? A more explicit analysis of dataset composition (object frequencies, prompt language patterns) could strengthen the paper, ensuring that the “diverse set” claim is well-supported and identifying any biases.
2. **None of the open-source models evaluated (including ImgEdit-E1) can actually perform multi-turn interactive editing** – they are all single-turn editors that don’t retain state. The paper evaluates multi-turn only on GPT-4o-Image and a mention of Gemini-2.0-Flash, since no other models support it. This is a limitation: the authors provide the data for multi-turn, but they do not present an open model trained specifically to use that data for iterative editing.
3. Can the authors provide more details on the human evaluation (how many raters, what criteria) and perhaps the extent of human agreement on the final leaderboard? Additionally, some qualitative examples in the paper (Figure 6) rely on visual inspection. It might help readers to know if humans clearly preferred ImgEdit-E1’s outputs over others.
4. Have the authors considered training an ImgEdit-based model that can handle multi-turn dialogues (for example, by incorporating a history or using an RNN/transformer to process a sequence of instructions)?
5. Can the authors clarify how much of ImgEdit-E1’s improvement comes from the new data versus the model architecture? For instance, how does Step1X-Edit (which uses a similar architecture) perform when trained on ImgEdit, if that was tried?

**Strengths Contributions:**

1. Quantitatively, ImgEdit has the highest GPT-4 quality score (4.71/5) among its prior datasets and the lowest “fake” detectability score (0.050, indicating very realistic edits), outperforming previous datasets on these metrics.
2. The dataset includes *multi-turn editing dialogues*, addressing scenarios requiring memory of past instructions, understanding implicit references, and undoing changes. The authors demonstrate that models trained on ImgEdit can handle these complex tasks (e.g. ImgEdit-E1 accurately performs multi-step edits and object extractions) which most baselines could not do.
3. The paper presents convincing evidence that the ImgEdit dataset improves model performance. ImgEdit-E1 (their trained model) consistently outperforms previous open-source editors like InstructPix2Pix, AnyEdit, UltraEdit’s model, etc., across almost all tasks on the benchmark.

---

> ### Author Rebuttal · Authors · 2025-07-30
>
> Thank you for the time, thorough comments, and nice suggestions. Your endorsement of our data quality and pipeline gives us significant encouragement. We are pleased to clarify your questions step-by-step.
>
> > **Q1:** The ImgEdit dataset’s diversity and realism are constrained by the underlying diffusion models, so a detailed analysis of its composition is needed to assess potential biases, including overrepresentation of certain styles or objects in prompt generation and filtering by GPT-4.
>
> **A1:** Thank you for the insightful question. We would like to clarify that not all edited images in ImgEdit are synthetic, with the exception that motion-change data is derived from different frames within videos(see lines 162–164). **We conducted analyses regarding potential biases in the stages of image generation, prompt generation, and filtering.**
>
> **Image Generation:**
> Potential biases may arise in the image editing process through the following mechanisms:
> * The shape of the mask can influence the generated result, including the object's shape and size.
> * Generative models inherently possess their own biases.
>
> Our pipeline accounts for these issues, either partially or entirely:
> * We experimented with enlarging the mask area or using bounding boxes that cover the mask region, and found that this approach yields the best results for removal and replacement tasks.
> * To further mitigate model-specific biases, our workflow utilizes multiple base models (SDXL, FLUX) as well as various community open-sourced LoRAs and Adapters.
>
> **Prompt Generation:**
> * Bias may also be introduced by the Large Language Model (LLM) during prompt construction. To address this, we utilized a more advanced LLM (GPT-4o). Table 1 in the manuscript demonstrates that our model captures a broader range of concepts in our data volume versus AnyEdit using LLaMA3-8B.
> * Additionally, we performed frequency analysis on the generated concepts and observed certain biases: ten terms (such as car, people, flowers, skin, etc.) accounted for 15% of the data. This issue is even more pronounced in other datasets, as evidenced by their word clouds.
>
> However, the uneven prompt distribution results from cumulative biases at various stages—including the content of the original images, bias in segmentation models, and bias in the LLM during prompt generation. Such biases are difficult to avoid. To mitigate this issue, semantic clustering methods could be used to filter out overrepresented concepts, or targeted data collection in underrepresented domains could be employed, allowing the LLM to generate prompts for these specific edits. For future work, it is crucial to incorporate sampling strategies that are sensitive to both important and long-tail editing classes and subjects.
>
> **Filtering Stage:**
> * We observed that GPT sometimes fails to accurately detect minor changes. Nevertheless, because we filtered out unobvious edited objects during pre-processing(line 159-160), this issue does not compromise the filtering accuracy.
>
> ---
>
> > **Q2:** The authors provide the data for multi-turn, but they do not present an open model trained specifically to use that data for iterative editing.
>
> **A2:** To assess the effectiveness of our multi-turn dataset, **we fine-tuned ImgEdit-E1 using multi-turn data**. We provide the text and visual encoders with context—including all previous images and instructions—during both training and inference to implement a basic version of multi-turn edit model.
>
> After training, we conducted a human evaluation to determine whether the model exhibits multi-turn capabilities. Given that the model is not exceptionally strong and current multi-turn editing implementation remains relatively naive, our model did not consistently succeed in every turn of the multi-turn tasks in ImgeditBench. This outcome was anticipated, as the results of gpt-4o-Image and gemini-2.5-flash on multi-turn ImgeditBench tasks were also unsatisfactory (detailed in Appendix D.3). Nevertheless, we observed that the model demonstrated certain multi-turn abilities. We performed human evaluation comparing the original model and the version fine-tuned on multi-turn data. Participants rated the models' ability to exhibit multi-turn competence. We recruited 10 participants, each of whom evaluated 10 randomly selected multi-turn editing tasks from multi-turn ImgeditBench. For each data point, the evaluator selected either a clear winner or indicated no preference.
>
> |   |Win| Tie| Lose|
> | ---| --- | --- | ---- |
> |ft v.s. w/o ft|36%|54%|10%|
>
> **The experimental results validate the effectiveness of multi-turn dataset** while also highlighting the limitations of the current architecture. Specifically, the DiT architecture suffers from semantic loss due to excessively long context—such as ignoring the results of previous editing rounds—which can lead to probabilistic failures in the editing process. Achieving stronger multi-turn editing capabilities may require architectures with enhanced history retention, longer context windows, or support for multi-image input, which we did not have sufficient time to explore during the rebuttal period. We will include this experiment results in the revision and hope that future architectures with better long-context support will enable more effective use of our dataset.
>
> ---
>
> > **Q3:** Can the authors provide more details on the human evaluation and perhaps the extent of human agreement on the final leaderboard? Also, It might help readers to know if humans clearly preferred ImgEdit-E1’s outputs over others.
>
> **A3:** **We conducted an additional human evaluation to compare the capabilities of ImgEdit-E1 with those of other editing models**, while also assessing the alignment between benchmark outcomes and human preferences. For this evaluation, we randomly selected five images from each subtask of ImgEditBench. For each original image and its corresponding editing instruction, edit results of each model were presented simultaneously to the evaluators. Considering the strong performance of gpt-4o-Image, we asked each evaluator to select the top two images from all candidates for each editing task. A total of ten evaluators participated in this assessment. Although the difference is modest, ImgEdit is generally preferred over Step1x-Edit and significantly outperforms all other models except gpt-4o-Image. In our revision, we will include this human
>  evaluation and additional visualization results in the appendix.
>
> |    |  Instruct-Pix2Pix | MagicBrush|AnySD| UltraEdit|ImgEdit-E1 |Step1X-Edit|GPT-4o-Image|
> | -- | -------- | ----  | ---- | --- | --- |--- | --- |
> | top2 pick ratio|0%|3%|5%|4%|51%|46%|91 %|
>
>
> Additionally, beyond the overall preference trends demonstrated by our human evaluation above, our benchmark shows patterns similar to other benchmarks such as GEditBench [1] and has been adopted in multiple published papers, indirectly reflecting the extent of human agreement on the final leaderboard.
>
> [1] Liu S, Han Y, Xing P, et al. Step1x-edit: A practical framework for general image editing[J]. arXiv preprint arXiv:2504.17761, 2025.
>
> ---
> > **Q4:** Can the authors clarify how much of ImgEdit-E1’s improvement comes from the new data versus the model architecture? For instance, how does Step1X-Edit (which uses a similar architecture) perform when trained on ImgEdit, if that was tried?
>
> **A4:** **We designed ablation experiments to verify that our improvements are attributable to the data rather than the model architecture.** Specifically, we kept the architecture of ImgEdit-E1 unchanged and ensured that all training hyperparameters remained consistent; detailed settings can be found in lines 789–797 of the Appendix. During training, we replaced all ImgEdit data with AnyEdit [1] data. The resulting model was then tested on ImgEditBench, the results are presented below. For comparison, we also included the results of AnySD.
>
> | sub-tasks  | AnySD | ImgEdit-E1 | ImgEdit-E1 with AnyEdit |
> |-------|----|-------|---------|
> | Remove | 2.23  | 2.40| 1.53  |
> | Background | 2.24  | 3.38| 2.55 |
> | Addition| 3.18  | 3.90 | 3.80  |
> | Style | 2.85  | 4.38  | 3.66  |
> | Adjust | 2.95  | 3.38 | 3.44 |
> | Extract  | 1.88  | 2.55| 2.02 |
> | Replace | 2.47  | 2.80  | 2.42|
> | Hybrid  | 1.56  | 2.87 | 1.78 |
> | Action  | 2.65  | 3.21 | 2.67 |
> |  **average**  | **2.45**  |  **3.21**  |  **2.76↓** |
>
> **The experimental results clearly demonstrate the superiority of the ImgEdit dataset over the previous state-of-the-art AnyEdit datasets.** We did not attempt to finetune Step1X-Edit, as it was pretrained on proprietary data and therefore cannot be fairly compared.
>
> [1] Yu Q, Chow W, Yue Z, et al. Anyedit: Mastering unified high-quality image editing for any idea[C]//Proceedings of the Computer Vision and Pattern Recognition Conference. 2025: 26125-26135.

---

> > ### Comment · Reviewer_vFmQ · 2025-08-01
> > **Response to Authors' Rebuttal**
> >
> > Thanks for your detailed response, which addressed most of my concerns. I will raise my rate.

---

> > > ### Author Response · Authors · 2025-08-01
> > >
> > > Dear Reviewer vFmQ,
> > >
> > > Thanks so much for your kind words and support. We really appreciate it! We’re glad our response addressed your concerns, and we’ll definitely carry forward your goodwill in future work and reviews.
> > >
> > > Best,
> > >
> > > On behalf of all ImgEdit authors

---

### Note · Authors · 2025-08-12

We sincerely thank the reviewers for their detailed and valuable comments, and we also appreciate the assistance from the area chair. **All reviewers (vFmQ, WnbF, YkpG, bQft) acknowledged the high quality and diversity of the dataset, and recognized the importance and novelty of the proposed editing types.** The comprehensive design of the benchmark and the consistency between model evaluation and human preferences were praised (bQft, YkpG). The clarity of motivation and writing was commended (YkpG), and the paper’s significant contributions to the image editing community were highlighted (WnbF).

Based on reviewers' valuable comments, we conclude some noteworthy replies for the reviewers, including:

- **[Reviewer vFmQ]** We further discussed the potential bias in the dataset and provided additional experimental validation.
- **[Reviewer vFmQ]** We conducted an ablation study on multi-turn data and verified its effectiveness.
- **[Reviewer vFmQ]** We supplemented our work with human evaluation to demonstrate the superiority of ImgEdit-E1.
- **[Reviewer YkpG, bQft]** We provided a detailed description of the human evaluation process for the Judge model (also see E.3 in the Supplementary Materials).
- **[Reviewer vFmQ, YkpG]** We added further ablation studies to once again demonstrate the superiority of ImgEdit data compared to previous datasets.
- **[Reviewer WnbF]** We increased the scale of human evaluation for the Judge model and obtained similar conclusions.
- **[Reviewer WnbF]** We explained the reasons for the fake score conflict between the dataset and model outputs.
- **[Reviewer YkpG]** We indicated the location of training and data pipeline details in the Supplementary Materials.
- **[Reviewer YkpG]** We clarified that there are no copyright issues in the manuscript.
- **[Reviewer bQft]** We provided experimental evidence for the consistency of edited data with respect to non-edited regions.
- **[Reviewer bQft]** We evaluated the training-free editing method suggested by the reviewer.
- **[Reviewer bQft]** We indicated the location of the benchmark details in paper.

**According to the reviewers‘ responses, We have already addressed all of the reviewers' concerns.**

---

### Decision · Program_Chairs · 2025-09-18

**Decision:**

Accept (poster)

**Comment:**

The authors introduced ImgEdit, a one-million-pair high-quality image-editing dataset, built using a multi-stage curation pipeline. They also propose ImgEdit-E1, a new editing model, and ImgEdit-Bench, a benchmark for evaluating instruction adherence, edit quality, and detail preservation. The evaluation of different models is also extensive. All the reviewers found the paper interesting and novel and the concerns raised during the rebuttal were addressed. The AC agrees with the positive recommendations and recommends acceptance.